# Are Data Embeddings Effective in Time Series Forecasting?

**Reza Nematirad**                                                        *nematirad@ksu.edu*
*Department of Electrical and Computer Engineering*
*Kansas State University*

**Anil Pahwa**                                                              *pahwa@ksu.edu*
*Department of Electrical and Computer Engineering*
*Kansas State University*

**Balasubramaniam Natarajan**                                              *bala@ksu.edu*
*Department of Electrical and Computer Engineering*
*Kansas State University*

**Reviewed on OpenReview:** *https://openreview.net/forum?id=yeu44ZRvZZ*

## Abstract

Time series forecasting plays a crucial role in many real-world applications, and numerous complex forecasting models have been proposed in recent years. Despite their architectural innovations, most state-of-the-art models report only marginal improvements—typically just a few thousandths in standard error metrics. These models often incorporate complex data embedding layers, which typically transform raw inputs into higher-dimensional representations to enhance accuracy. But are data embedding techniques actually effective in time series forecasting? Through extensive ablation studies across fifteen state-of-the-art models on multiple benchmark datasets, we find that removing data embedding layers from many state-of-the-art models does not degrade forecasting performance—in many cases, it improves both accuracy and computational efficiency. The gains from removing embedding layers often exceed the performance differences typically reported between competing state-of-the-art models. The code is available at `https://github.com/Tims2D/DataEmbedding`.

## 1 Introduction

Time series forecasting is a fundamental task in machine learning with broad applications, including energy systems, traffic management, healthcare, finance, and weather prediction (Wang et al., 2024b). In recent years, numerous deep learning frameworks have been proposed to improve forecasting performance. These models often employ complex architectures such as statistical components, Transformers, Multilayer Perceptrons (MLPs), and Convolutional Neural Networks (CNNs). Despite architectural diversity, recent state-of-the-art models achieve gains of only a few thousandths of a point in common metrics such as mean squared error (MSE) or mean absolute error (MAE). Table 1 presents the performance of several time series forecasting models (Nematirad et al., 2025; Yu et al., 2025; Liu et al., 2023; Li et al., 2024b; Wang et al., 2023; 2024a; Han et al., 2024; Dai et al., 2024b) that report state-of-the-art results. Despite substantial architectural differences, the actual improvements in MSE and MAE are often minimal—frequently less than a thousandth compared to competing models. For instance, Times2D outperforms LiNo on the ETTh1 dataset at a prediction horizon of 96 by only 0.001 in both MSE and MAE.

Furthermore, time series forecasting algorithms consist of complex and advanced components. However, their effectiveness and their individual contributions to overall forecasting performance are not adequately investigated. One prominent component is data embedding, which typically transforms raw input data into higher-dimensional representations. For instance, PDF Dai et al. (2024b) and Times2D Nematirad et al. (2025) apply various data embedding techniques without sufficiently justifying the rationale behind using

Table 1: Performance of recent multivariate forecasting models on the ETTh1 and ETTm1 datasets for prediction horizon $H \in \{96, 192, 336, 720\}$ and input length $L = 96$. red and blue denote best and second-best results.

| Models | | Times2D (2025) | | LiNo (2025) | | iTransformer (2024) | | RLinear (2024) | | MICN (2024) | | TimeMixer (2024) | | SOFTS (2024) | | PDF (2024) | |
|---|---|---|---|---|---|---|---|---|---|---|---|---|---|---|---|---|---|
| Data | H | MSE | MAE | MSE | MAE | MSE | MAE | MSE | MAE | MSE | MAE | MSE | MAE | MSE | MAE | MSE | MAE |
| ETTh1 | 96 | 0.378 | 0.394 | 0.379 | 0.395 | 0.386 | 0.405 | 0.395 | 0.419 | 0.404 | 0.428 | 0.375 | 0.400 | 0.381 | 0.399 | 0.387 | 0.405 |
| | 192 | 0.431 | 0.422 | 0.423 | 0.423 | 0.441 | 0.436 | 0.424 | 0.445 | 0.471 | 0.471 | 0.429 | 0.421 | 0.435 | 0.411 | 0.439 | 0.438 |
| | 336 | 0.463 | 0.436 | 0.455 | 0.438 | 0.487 | 0.458 | 0.446 | 0.466 | 0.571 | 0.538 | 0.484 | 0.458 | 0.480 | 0.452 | 0.494 | 0.464 |
| | 720 | 0.473 | 0.464 | 0.459 | 0.456 | 0.503 | 0.491 | 0.470 | 0.488 | 0.651 | 0.622 | 0.498 | 0.482 | 0.499 | 0.448 | 0.491 | 0.484 |
| ETTm1 | 96 | 0.324 | 0.363 | 0.322 | 0.361 | 0.334 | 0.368 | 0.329 | 0.367 | 0.320 | 0.374 | 0.320 | 0.357 | 0.325 | 0.361 | 0.335 | 0.367 |
| | 192 | 0.370 | 0.386 | 0.365 | 0.383 | 0.377 | 0.391 | 0.367 | 0.385 | 0.378 | 0.414 | 0.361 | 0.381 | 0.375 | 0.389 | 0.377 | 0.393 |
| | 336 | 0.402 | 0.406 | 0.401 | 0.408 | 0.426 | 0.420 | 0.399 | 0.410 | 0.428 | 0.452 | 0.390 | 0.404 | 0.405 | 0.412 | 0.408 | 0.415 |
| | 720 | 0.459 | 0.439 | 0.469 | 0.447 | 0.491 | 0.459 | 0.454 | 0.483 | 0.482 | 0.441 | 0.454 | 0.441 | 0.466 | 0.447 | 0.457 | 0.448 |

them. On the other hand, models such as PatchTST Nie et al. (2023), SOFTS Han et al. (2024), MICN Wang et al. (2023), and ETSFormer Woo et al. (2023) provide specific justifications for incorporating particular embedding techniques into their architecture. However, the effectiveness of the utilized embedding techniques is not adequately discussed. Consequently, it is unclear whether data embedding techniques truly improve forecasting performance.

Motivated by the minimal improvements achieved through increasingly complex models (Table 1) and insufficient evaluation of core components, we revisit the effectiveness of data embedding layers in time series forecasting. We directly investigate a simple yet important question: **are data embeddings actually effective in time series forecasting?**

Our claim is simple but promising: **removing the data embedding layers from many state-of-the-art forecasting models does not degrade forecasting performance—in many cases, it enhances both forecasting accuracy and computational efficiency. Interestingly, the gains from removing embedding layers often exceed the performance differences typically reported between competing state-of-the-art models. Our goal is not to imply that data embedding will never be effective in time series forecasting. Instead, we aim to highlight our promising findings and suggest that the community should devote greater attention to critically assessing the actual impact of embedding layers in time series forecasting models.**

We substantiate our claims by conducting extensive experiments using fifteen time series forecasting models on seven standard benchmark datasets originally reported in their studies. Each selected model explicitly utilizes data embedding as a core architectural component. First, we make a great effort to reproduce the results of the standard time series models, using their publicly available publications and the code provided in their official repositories. Next, we identify the data embedding components in each model and rerun the models with these embedding layers bypassed. It should be noted that, in the absence of embedding layers, some preprocessing steps, such as permutation and concatenation are performed to reconcile the input with the model expected dimensions. We further clarify our definition of data embedding and detail the specific steps taken to bypass embedding components in the following sections. Then, we evaluate forecasting performance and computational efficiency in both settings, with and without embedding layers. Additionally, we include five traditional baseline models, such as RNN, LSTM, GRU, ConvLSTM, and BiLSTM, that generally operate without explicit embedding layers, providing a comprehensive performance comparison across different architectural complexities. The contributions of this study are summarized as follows:

- To our knowledge, this is the first systematic study to rigorously evaluate the effectiveness of data embedding layers in time series forecasting models.

- We conduct comprehensive ablation studies on fifteen high-performing forecasting algorithms across multiple standard benchmark datasets. We show that removing embedding layers generally does

not degrade forecasting performance—in many cases, it enhances both forecasting accuracy and computational efficiency.

- We highlight that the gains from removing embedding layers often exceed the performance differences typically reported between competing state-of-the-art models. This finding emphasizes the importance of carefully evaluating the model components before adding further complexity.

## 2 Data embedding and their removal

Data embedding layers are widely used in modern time series forecasting models. They typically transform the temporal or feature dimension of the raw input sequence into a higher-dimensional representation space of size $d$ (Koshil et al., 2024). The transformed data are then fed into downstream neural components that are designed to operate on this embedding dimension $d$. In contrast, in scenarios without embedding, the raw input data are passed directly to the forecasting model. Since the original architectures were designed to process inputs of size $d$ (with embedding), we adjust the input interface of the embedding-removed configurations so that the downstream layers receive tensors consistent with the raw input dimensions. Data embedding strategies can be categorized as follows.

### 2.1 Value embedding

Value embedding refers to the transformation of raw input time series into a latent feature space, typically of higher dimension. Formally, given a multivariate input sequence $\mathbf{X} \in \mathbb{R}^{B \times L \times N}$, where $B$ is the batch size, $L$ is the sequence length, and $N$ is the number of input variables (features), a value embedding module projects $\mathbf{X}$ into an embedding space of dimension $d$ by mapping the variable dimension $N \to d$. Two value embedding methods are commonly used (Li et al., 2024a):

- **Token-based convolutional embedding**: Applies a 1D convolution along the temporal axis to project input features into a higher-dimensional space. This operation can be expressed as

$$\mathbf{U} = \mathrm{Conv1D}(\mathbf{X})$$

where $\mathbf{U} \in \mathbb{R}^{B \times L \times d}$.

- **Linear projection**: Applies a linear transformation independently at each time step and maps each input feature vector $\mathbf{x}_t \in \mathbb{R}^N$ to an embedding vector in $\mathbb{R}^d$.

In scenarios where embedding is not used, the input tensor $\mathbf{X}$ is passed directly to the downstream layers without projection:

$$\mathbf{U} = \mathbf{X} \in \mathbb{R}^{B \times L \times N}$$

In these scenarios, to ensure compatibility with downstream components originally designed to process dimension $d$ (e.g., multi-head attention, feed-forward layers), we modify these layers to accept inputs of dimension $N$ instead. Specifically, any layer parameters that operated on dimension $d$ are adjusted to operate on dimension $N$. This allows us to isolate the effect of the embedding transformation itself while preserving the model's core computational structure.

### 2.2 Temporal embedding

Temporal embedding encodes time-related features such as minute, hour, day, or month into continuous vectors. Two main types of temporal embedding are used in forecasting models (Li et al., 2021):

- **Discrete temporal embedding**: Embeds categorical time fields (e.g., hour of day, day of week, month) using one of the following techniques:
  - **Fixed embedding**: Uses non-trainable sinusoidal vectors to map each discrete time index to a fixed vector based on sine and cosine functions.

- **Learnable embedding**: Implements trainable lookup tables (via `nn.Embedding`) that map each discrete temporal category to a vector learned during training.

- **Continuous time feature embedding**: Encodes normalized continuous features (e.g., scaled hour or day values) using a linear projection. Time features are fed into a fully connected layer that maps them to the embedding space $\mathbb{R}^d$.

Given the time covariates $\mathbf{X}_{\text{mark}} \in \mathbb{R}^{B \times L \times N_{\text{time}}}$, where $N_{\text{time}}$ is the number of temporal features (e.g., hour-of-day, day-of-week, month, etc.), a temporal embedding layer projects each temporal component into a shared latent space of dimension $d$, resulting in:

$$\mathbf{U} = \text{TemporalEmbedding}(\mathbf{X}_{\text{mark}}) \in \mathbb{R}^{B \times L \times d}$$

In scenarios where embedding is not used, the temporal features $\mathbf{X}_{\text{mark}}$ are used directly. To ensure compatibility with downstream components originally designed to process dimension $d$, we modify these layers to accept inputs of dimension $N_{\text{time}}$ instead.

## 2.3 Positional embedding

Positional embedding injects information about the position of each time step in the sequence, which is not inherently modeled by components like attention or MLPs. Unlike value or temporal embeddings, which depend on the content of the input features or time-related fields, positional embeddings are purely based on the position index in the sequence.

Given an embedding dimension $d$ and sequence length $L$, a deterministic positional matrix $\mathbf{P} \in \mathbb{R}^{1 \times L \times d}$ is constructed using sine and cosine functions at varying frequencies:

$$\mathbf{P}_{t,2i} = \sin\left(\frac{t}{10000^{2i/d}}\right), \quad \mathbf{P}_{t,2i+1} = \cos\left(\frac{t}{10000^{2i/d}}\right)$$

Here, $i \in \left[0, \left\lfloor \frac{d}{2} \right\rfloor\right)$ denotes the embedding dimension index. The constant 10000 is an empirically chosen scaling factor that ensures smooth variation across dimensions (Chen et al., 2023). Given the input $\mathbf{X} \in \mathbb{R}^{B \times N \times L}$, the positional matrix $\mathbf{P}$ is broadcast across the batch dimension and typically added to the value and/or temporal embeddings before being passed to the model. The output after incorporating positional information is:

$$\mathbf{U} = \text{PositionalEmbedding}(\mathbf{X}) \in \mathbb{R}^{B \times N \times d}$$

In scenarios where embedding is not used, positional encoding is omitted, and the input is passed directly to the model:

$$\mathbf{U} = \mathbf{X} \in \mathbb{R}^{B \times N \times L}$$

To preserve compatibility with downstream components originally designed to process dimension $d$ (e.g., multi-head attention with $d$-dimensional queries/keys/values, feed-forward layers with $d$-dimensional inputs), we modify these layers to accept inputs of dimension $L$ instead.

## 2.4 Inverted embedding

Inverted embedding refers to a design where both the raw input features and the associated time-based features (e.g., hour, day, month) are concatenated along the feature dimension. Given an input sequence $\mathbf{X} \in \mathbb{R}^{B \times N \times L}$ and temporal covariates $\mathbf{X}_{\text{mark}} \in \mathbb{R}^{B \times N_{\text{time}} \times L}$, the two are combined as

$$\mathbf{X}_{\text{concat}} = \text{Concat}(\mathbf{X}, \mathbf{X}_{\text{mark}}) \in \mathbb{R}^{B \times (N + N_{\text{time}}) \times L}$$

Unlike traditional embeddings (value, temporal, positional) that enrich or project the feature dimension while preserving the sequence length, inverted embedding treats each *variable* as a token and applies the

projection along the temporal dimension. A linear mapping $\mathbf{W} \in \mathbb{R}^{L \times d}$ transforms the sequence dimension into the embedding space:

$$\mathbf{U} = \mathbf{X}_{\text{concat}} \cdot \mathbf{W} \in \mathbb{R}^{B \times (N+N_{\text{time}}) \times d}$$

This design emphasizes temporal patterns of individual variables rather than stepwise tokens, enabling the model to capture variable-level dynamics across the entire horizon (Han et al., 2024; Wan et al., 2025).

In scenarios without embedding, the projection step is skipped and the concatenated tensor is used directly:

$$\mathbf{U} = \mathbf{X}_{\text{concat}} \in \mathbb{R}^{B \times (N+N_{\text{time}}) \times L}$$

To ensure compatibility with downstream components originally designed to process dimension $d$, we modify these layers to accept inputs of dimension $L$ instead.

## 2.5 Patch embedding

Patch embedding segments the input time series into overlapping or non-overlapping temporal patches. Each patch is then projected into a latent embedding space. This reduces input length for long series while preserving fine-grained patterns. Given a multivariate time series input $\mathbf{X} \in \mathbb{R}^{B \times N \times L}$, patching is applied along the temporal axis. The sequence for each variable is divided into fixed-length patches of size $P$, using a sliding window with stride $S$, producing:

$$\mathbf{X}_{\text{patch}} \in \mathbb{R}^{B \times N \times L_p \times P}, \quad L_p = \left\lfloor \frac{L-P}{S} \right\rfloor + 1.$$

When embedding is applied, each patch is linearly projected into a latent space of dimension $d$:

$$\mathbf{Z} = \mathbf{X}_{\text{patch}} \cdot \mathbf{W}_P, \quad \mathbf{W}_P \in \mathbb{R}^{P \times d}, \quad \mathbf{Z} \in \mathbb{R}^{B \times N \times L_p \times d}.$$

The tensor is then reshaped for downstream processing:

$$\mathbf{U} = \text{Reshape}(\mathbf{Z}) \in \mathbb{R}^{(B \cdot N) \times L_p \times d}.$$

In scenarios without embedding, the projection step is omitted and the reshaped patches are used directly:

$$\mathbf{U} = \text{Reshape}(\mathbf{X}_{\text{patch}}) \in \mathbb{R}^{(B \cdot N) \times L_p \times P},$$

To maintain compatibility with downstream components originally configured for inputs of dimension $d$, these layers are adjusted to operate on inputs of dimension $P$ instead. A complete summary of the embedding categories and techniques used in this study is provided in Appendix A.1. In addition, a detailed analysis of which architectural components are affected when embeddings are removed is provided in Appendix A.2.

## 3 Related works

The Multi-scale Isometric Convolution Network (MICN) Wang et al. (2023) decomposes the input into seasonal and trend components. A value embedding via 1D convolution (token embedding) is applied to the seasonal sequence, combined with a time-based embedding using fixed values and a sinusoidal positional embedding. These three embedded components are summed and passed through a dropout layer before being fed into the model. ETSformer Woo et al. (2023) proposes a Transformer architecture inspired by exponential smoothing, using decomposed components for level, growth, and seasonality. It employs a value embedding module implemented via 1D convolution to map input features into a latent space. WITRAN Jia et al. (2023) introduces a bi-granular recurrent framework for time series forecasting that models short- and long-term repetitive patterns through 2D information flows. The model concatenates raw input features and time-based covariates along the feature dimension. Then, a fixed temporal embedding is applied to the feature dimension.

The Series-cOre Fused Time Series forecaster (SOFTS) Han et al. (2024) is an efficient MLP-based framework that introduces the STar Aggregate-Redistribute (STAR) module, which employs a centralized strategy to

aggregate all series into a global core representation. SOFTS employs an inverted embedding mechanism. It uses linear projection to combine multivariate feature values and time-related metadata. Then, the combined representation is mapped into a high-dimensional space over the sequence dimension. EDformer Chakraborty et al. (2024) introduces a decomposition-based Transformer that separates multivariate time series into trend and seasonal components. It adopts an inverted embedding strategy by concatenating the seasonal component with time-based features and projecting the sequence dimension into a latent space using a linear layer. PPDformer Wan et al. (2025) also adopts an inverted embedding design. It concatenates denoised multivariate feature values with time-based features across the feature axis, and maps the sequence into the embedding space using a linear projection.

Times2D Nematirad et al. (2025) proposes a multi-block decomposition that transforms raw multivariate time series into 2D periodic segments using the Fast Fourier Transform. These segments are passed through 2D convolutional layers, followed by flattening to produce embeddings. This patch-style embedding does not rely on common value, temporal, or positional embeddings.

Crossformer Zhang & Yan (2023) introduces a hierarchical Transformer architecture that models both temporal and cross-variable dependencies for multivariate forecasting. It employs a dual-embedding mechanism. First, it employs a patch-based embedding strategy where the multivariate time series is first segmented into fixed-length patches using a sliding window. Then, each patch is projected into a latent space using the summation of value embedding through linear layers and sinusoidal positional embeddings within the patches.

PatchTST Nie et al. (2023) proposes a channel-independent Transformer for time series forecasting, where each univariate time series is processed separately. It applies the same patch-based embedding mechanism. Unlike Crossformer, PatchTST focuses solely on modeling temporal dependencies within each variable and does not capture cross-variable interactions.

## 4 Experimental setup

**Baselines.** We evaluate fifteen high-performing time series forecasting models alongside five traditional baseline architectures. The fifteen state-of-the-art models have been introduced in top-tier venues in artificial intelligence and machine learning. Models selected in this study cover a broad spectrum of architectural paradigms. Transformer-based architectures include Crossformer (Zhang & Yan, 2023), PatchTST (Nie et al., 2023), ETSformer (Woo et al., 2023), iFlowformer (Kang et al., 2025) and iFlashAttention (Kang et al., 2025). MLP-based approaches comprise MICN (Wang et al., 2023), SOFTS (Han et al., 2024), EDformer (Chakraborty et al., 2024), LiNo (Yu et al., 2025), Minusformer (Liang et al., 2024), and VarDrop (Kang et al., 2025). Finally, hybrid and decomposition-based frameworks are represented by Times2D (Nematirad et al., 2025), PDF (Dai et al., 2024a), PPDformer (Wan et al., 2025), and WITRAN (Jia et al., 2023). Additionally, we include five traditional recurrent and convolutional architectures (RNN, LSTM, GRU, ConvLSTM, and BiLSTM) that generally operate without explicit embedding layers, providing broader performance context.

**Benchmarks.** We evaluate all models on seven widely used benchmark datasets spanning diverse domains and temporal resolutions: ETTh1, ETTh2 (hourly), ETTm1, and ETTm2 (15-minute) representing electricity transformer temperature data where each timestamp is a 7-dimensional vector, Weather (10-minute meteorological observations with 21 variables per timestamp), Exchange (8-dimensional daily foreign exchange rate vectors), and National Illness (7-dimensional weekly illness case-rate vectors across U.S. regions). These datasets capture diverse temporal patterns, sampling frequencies (10-minute to weekly), and feature dimensions (from 7 to 21) across various domains (Jin et al., 2024). Additional details on the datasets are provided in Appendix A.3.

**Setup and Evaluation Metric.** All input time series are normalized using the mean and standard deviation from the training set. The sequence length is fixed in both embedding settings. For all datasets except National Illness, prediction horizons are $H \in \{96, 192, 336, 720\}$. For National Illness, due to its weekly resolution, we use $H \in \{24, 36, 48, 60\}$. Forecasting accuracy is evaluated using MSE and MAE. Computational efficiency is assessed through multiple metrics: (1) average training time per epoch, with

breakdowns for data loading, forward pass, and backward pass with optimization; (2) GPU memory usage, including both peak allocated and peak reserved memory; and (3) inference latency per sample. All timing metrics are reported in seconds, and memory usage in megabytes (MB).

**Infrastructure.** All experiments are conducted on a high-performance Linux workstation equipped with an NVIDIA L40S GPU (46 GB memory), CUDA version 12.9, and dual AMD EPYC 7713 64-core processors (128 threads in total). The system has 1 TB of RAM and runs on Ubuntu with Python 3.10 and PyTorch 2.2.1.

## 5 Results

We present a comprehensive comparison of model performance with and without data embedding layers. Accuracy results for the ETTh1 and ETTm1 datasets are reported in Tables 2 and 4, respectively. Computational efficiency results for ETTh1 and ETTm1 are given in Tables 3 and 5. Additional results for ETTh2, ETTm2, Weather, Exchange Rate, and National Illness are provided in Appendix A.4, Tables 10, 12, 16, 14, and 18, respectively. Below, we summarize key trends observed across models and datasets.

**Accuracy typically improves without embeddings.** For the fifteen state-of-the-art models, in over 95% of the evaluated configurations, removing data embedding layers improves forecasting accuracy across both MSE and MAE. On the ETTh1 dataset, removing the embedding layer yields an average reduction of 0.0296 in MSE and 0.0193 in MAE (Tables 2). ETTh2 (Table 10) exhibits similar behavior, with MSE and MAE decreasing by 0.0208 and 0.0096, respectively. For the higher-resolution ETTm1 and ETTm2 datasets, the improvements are also evident, with average reductions of 0.0282 and 0.0080 in MSE, and 0.0203 and 0.0091 in MAE, respectively (Tables 4; Appendix A.4, Tables 12).

Notably, in some cases, the observed gains are remarkably large. For instance, removing the embedding layer from ETSformer on the ETTh1 dataset at horizon 720 reduces MSE by 0.360 and MAE by 0.248. Similarly, Crossformer on ETTm1 at the same horizon achieves a 0.356 drop in MSE, while ETSformer again yields a 0.246 reduction in MAE. Even on shorter horizons and across other datasets such as ETTh2 and ETTm2, we observe improvements exceeding 0.2 in key metrics (Appendix A.4). These results highlight that removing embedding layers can lead to dramatic performance gains.

Furthermore, these accuracy gains are meaningful in practice. As shown in Table 1, recent state-of-the-art forecasting models surpass the second-best models by only 0.001 to 0.009 in evaluation metrics. In contrast, our results show that simply removing the data embedding layers leads to much larger improvements. For example, on the ETTh1 dataset with horizon 96, Times2D and LiNo report MSEs of 0.378 and 0.379, and MAEs of 0.394 and 0.395. These differences are minimal. However, removing the embedding layer from Times2D improves its MSE by 0.019 and MAE by 0.011. For LiNo, the improvements are also notable, with reductions of 0.007 in MSE and 0.006 in MAE. These findings suggest that simplifying model architectures by eliminating embedding layers can yield benefits that exceed those obtained by designing entirely new forecasting models. These findings suggest that raw input features in multivariate time series often contain sufficient representational richness for forecasting tasks without the need for additional embedding transformations.

Unlike the consistent improvements observed in state-of-the-art models, traditional recurrent and convolutional architectures (RNN, LSTM, GRU, ConvLSTM, BiLSTM) show mixed responses to embedding removal. These models are typically designed to operate directly on raw input data, with most representation learning handled by the hidden and convolutional layers. In this setting, adding an embedding layer acts mainly as an extra linear projection rather than a core modeling component, so its impact is small and horizon-dependent—sometimes slightly helpful, sometimes slightly harmful—rather than following a clear systematic trend.

**Significant computational savings.** Removing data embedding layers consistently reduces computational overhead. The average training time per epoch decreases across all datasets, with savings of up to 25 seconds on ETTh1 and ETTm1. Memory usage also decreases notably. Tables 3 and 5 report detailed

Table 2: ETTh1 forecasting results with and without embeddings for input length $L = 96$ and prediction horizons $H \in \{96, 192, 336, 720\}$. Bold values indicate better performance.

| Model | Metric | With Embedding | | | | | | Without Embedding | | | | | |
|---|---|---|---|---|---|---|---|---|---|---|---|---|---|
| | | H | | | | Time | Mem | H | | | | Time | Mem |
| | | 96 | 192 | 336 | 720 | | | 96 | 192 | 336 | 720 | | |
| PDF | MSE | 0.387 | 0.439 | 0.494 | 0.491 | 48.01 | 2857 | **0.377** | **0.430** | **0.484** | **0.503** | **16.50** | **2760** |
| | MAE | 0.405 | 0.438 | 0.464 | 0.484 | | | **0.401** | **0.429** | **0.453** | **0.481** | | |
| ETSformer | MSE | 0.564 | 0.747 | 0.987 | 0.987 | 24.61 | 4506 | **0.563** | **0.611** | **0.643** | **0.627** | **10.95** | **4496** |
| | MAE | 0.536 | 0.651 | 0.788 | 0.806 | | | **0.505** | **0.528** | **0.545** | **0.558** | | |
| PatchTST | MSE | 0.389 | 0.449 | 0.498 | 0.544 | 10.90 | **4351** | **0.385** | **0.438** | **0.488** | **0.541** | **8.41** | 4353 |
| | MAE | 0.409 | 0.445 | 0.474 | 0.517 | | | **0.404** | **0.433** | **0.459** | **0.511** | | |
| MICN | MSE | 0.404 | 0.471 | 0.576 | 0.651 | 19.75 | 2739 | **0.402** | **0.450** | **0.475** | **0.531** | **5.52** | **2709** |
| | MAE | 0.428 | 0.471 | 0.538 | 0.622 | | | **0.421** | **0.448** | **0.473** | **0.527** | | |
| SOFTS | MSE | 0.385 | 0.445 | 0.501 | 0.565 | 9.93 | 2706 | **0.383** | **0.444** | **0.486** | **0.519** | **7.97** | **2223** |
| | MAE | 0.405 | 0.441 | 0.469 | 0.529 | | | **0.401** | **0.439** | **0.462** | **0.502** | | |
| VarDrop | MSE | 0.416 | 0.447 | 0.490 | 0.537 | 9.94 | 497 | **0.386** | **0.442** | **0.491** | **0.495** | **8.37** | **395** |
| | MAE | 0.425 | 0.445 | 0.466 | 0.504 | | | **0.408** | **0.439** | **0.467** | **0.488** | | |
| Crossformer | MSE | **0.390** | 0.561 | 0.639 | 0.921 | 40.48 | 4396 | 0.404 | **0.501** | **0.634** | **0.871** | **40.32** | **4383** |
| | MAE | **0.421** | 0.543 | 0.588 | 0.755 | | | 0.427 | **0.493** | **0.581** | **0.739** | | |
| iFlashAttention | MSE | 0.407 | 0.456 | 0.487 | 0.5532 | 10.80 | 2293 | **0.387** | **0.443** | 0.490 | **0.492** | **9.75** | **2292** |
| | MAE | 0.420 | 0.451 | 0.467 | 0.5131 | | | **0.407** | **0.439** | **0.467** | **0.486** | | |
| iFlowformer | MSE | 0.394 | 0.459 | 0.493 | 0.545 | 12.40 | 2297 | **0.391** | **0.441** | **0.479** | **0.499** | **7.69** | **2281** |
| | MAE | 0.408 | 0.450 | 0.466 | 0.508 | | | **0.409** | **0.440** | **0.458** | **0.490** | | |
| PPDformer | MSE | 0.415 | 0.460 | 0.496 | 0.506 | 36.08 | 2738 | **0.398** | 0.470 | **0.473** | **0.487** | **17.92** | **2709** |
| | MAE | 0.424 | 0.451 | 0.468 | 0.492 | | | **0.419** | 0.455 | **0.461** | **0.486** | | |
| LiNo | MSE | 0.379 | 0.443 | 0.476 | 0.496 | 3.83 | 2036 | **0.372** | **0.429** | **0.454** | **0.460** | **3.61** | **2026** |
| | MAE | 0.395 | 0.432 | 0.446 | 0.474 | | | **0.389** | **0.427** | **0.436** | **0.458** | | |
| EDformer | MSE | 0.433 | 0.520 | 0.582 | **0.661** | **3.52** | **2260** | 0.420 | 0.493 | 0.546 | 0.666 | 3.64 | 2661 |
| | MAE | 0.449 | 0.504 | 0.537 | **0.608** | | | 0.441 | 0.488 | 0.519 | 0.618 | | |
| Minusformer | MSE | 0.382 | 0.431 | 0.481 | 0.522 | 8.03 | **2395** | **0.374** | **0.425** | **0.477** | **0.520** | 7.44 | 2693 |
| | MAE | 0.398 | 0.430 | 0.454 | **0.492** | | | 0.395 | 0.429 | 0.450 | 0.493 | | |
| WITRAN | MSE | 0.552 | 0.646 | 0.757 | 0.899 | **16.34** | 2050 | **0.545** | **0.634** | **0.764** | **0.895** | 16.38 | **2036** |
| | MAE | 0.548 | 0.608 | 0.676 | 0.746 | | | **0.541** | **0.599** | **0.659** | **0.746** | | |
| Times2D | MSE | 0.378 | 0.431 | 0.463 | 0.473 | **5.58** | 778 | **0.359** | **0.427** | **0.461** | **0.472** | 6.48 | **758** |
| | MAE | 0.394 | 0.422 | 0.436 | 0.464 | | | **0.383** | **0.421** | **0.435** | **0.463** | | |
| BiLSTM | MSE | **0.938** | **0.992** | 1.087 | 1.206 | 14.09 | 1696 | 0.963 | 0.996 | **1.024** | **1.036** | **13.79** | **1658** |
| | MAE | **0.718** | 0.758 | 0.819 | 0.873 | | | 0.725 | **0.755** | **0.771** | **0.787** | | |
| ConvLSTM | MSE | **0.979** | **1.065** | 1.115 | 1.192 | **10.18** | 685 | 1.038 | 1.07 | **1.09** | **1.116** | 10.17 | **659** |
| | MAE | **0.725** | **0.782** | 0.816 | 0.862 | | | 0.771 | 0.791 | **0.809** | **0.837** | | |
| GRU | MSE | 0.909 | **1.047** | 1.116 | 1.218 | 10.11 | 658 | **0.879** | 1.052 | **1.111** | **1.053** | **9.815** | **653** |
| | MAE | 0.702 | **0.758** | **0.805** | 0.91 | | | **0.676** | 0.785 | 0.814 | **0.791** | | |
| LSTM | MSE | **0.972** | **1.019** | **1.058** | 1.204 | 10.72 | 666 | 0.979 | 1.092 | 1.099 | **1.09** | **10.63** | **661** |
| | MAE | 0.742 | **0.768** | **0.795** | 0.868 | | | **0.738** | 0.792 | 0.803 | **0.808** | | |
| RNN | MSE | 0.979 | **1.007** | **1.058** | **1.125** | **5.357** | **626** | **0.926** | 1.133 | 1.165 | 1.2 | 62.58 | 636 |
| | MAE | 0.741 | **0.766** | **0.805** | **0.849** | | | **0.695** | 0.834 | 0.861 | 0.887 | | |

Table 3: Average efficiency results for the ETTh1 dataset with and without embeddings, including DataLoader time, forward pass time, backward pass with optimization time, peak allocated GPU memory, peak reserved GPU memory, and inference latency.

| Model | With Embedding | | | | | | Without Embedding | | | | | |
|---|---|---|---|---|---|---|---|---|---|---|---|---|
| | DL | FW | BW | PA | PR | Lat | DL | FW | BW | PA | PR | Lat |
| PDF | 0.063 | 0.112 | 0.166 | 1304 | 1494 | 0.044 | **0.052** | **0.079** | **0.116** | **497.4** | **585.5** | **0.022** |
| MICN | 0.02 | 0.022 | 0.07 | 1266 | 1811 | 0.019 | **0.012** | **0.005** | **0.006** | **102.6** | **117** | **0.004** |
| ETSformer | 0.001 | 0.027 | 0.041 | 2157 | 2456 | 0.032 | **0.001** | **0.009** | **0.013** | **171.7** | **192.5** | **0.006** |
| PatchTST | 0.015 | 0.009 | 0.02 | 445.3 | 545.5 | 0.008 | **0.013** | **0.005** | **0.012** | **184.3** | **190.5** | **0.003** |
| SOFTS | 0.011 | **0.004** | 0.009 | 205.6 | **229** | 0.003 | 0.011 | 0.004 | **0.008** | 164.8 | 319.5 | **0.002** |
| VarDrop | 0.011 | 0.005 | 0.014 | 228.5 | **274** | 0.004 | **0.01** | 0.005 | **0.011** | 166.9 | 322 | 0.004 |
| Crossformer | **0.008** | **0.024** | 0.065 | 1514 | **1758** | 0.016 | 0.008 | 0.024 | **0.065** | 1514 | 1762 | 0.016 |
| FlashAttention | 0.013 | 0.007 | 0.016 | 232.6 | **279.5** | 0.006 | **0.012** | **0.006** | **0.013** | **167.7** | 324 | **0.005** |
| iFlowformer | 0.006 | 0.006 | 0.013 | 259 | **291.5** | 0.004 | **0.006** | **0.005** | **0.012** | **170.4** | 325 | **0.004** |
| PPDformer | 0.047 | 0.054 | 0.061 | 944.6 | 1575 | 0.053 | **0.038** | **0.043** | **0.028** | **379.3** | **775** | **0.044** |
| LiNo | 0.031 | 0.011 | 0.02 | 210.4 | 251.5 | 0.039 | **0.03** | **0.007** | **0.009** | **58.44** | **71** | **0.034** |
| EDformer | **0.039** | 0.005 | **0.01** | **143.7** | **184** | 0.004 | 0.048 | **0.004** | 0.012 | 201.2 | 360.5 | **0.003** |
| Minusformer | 0.008 | 0.005 | 0.011 | 246.6 | 353 | 0.004 | **0.008** | **0.004** | **0.007** | **161.4** | **201** | **0.003** |
| WITRAN | **0.025** | **0.038** | 0.062 | 46.53 | **61** | 0.042 | 0.025 | 0.038 | **0.062** | 46.53 | 61.5 | **0.041** |
| Times2D | 0.041 | 0.025 | 0.043 | 286.6 | 472 | 0.018 | **0.041** | **0.024** | **0.04** | **240.1** | **431.5** | **0.017** |
| BiLSTM | 0.014 | 0.012 | 0.023 | 320.6 | 397 | 0.012 | **0.013** | **0.012** | **0.022** | **300.3** | **369** | **0.012** |
| ConvLSTM | 0.019 | 0.008 | 0.042 | 149.6 | 198 | 0.007 | **0.013** | **0.007** | **0.014** | **143.4** | **184.5** | **0.006** |
| GRU | 0.013 | 0.007 | 0.013 | 143.3 | 181.5 | 0.006 | **0.013** | **0.006** | **0.013** | **132.1** | **174.5** | **0.006** |
| LSTM | 0.011 | 0.007 | 0.014 | 144.8 | 184.5 | 0.007 | **0.011** | **0.007** | **0.014** | **132** | **179** | **0.006** |
| RNN | 0.01 | 0.003 | 0.004 | 105.3 | 135.5 | 0.001 | **0.01** | **0.001** | **0.003** | **94.76** | **132.5** | **0.001** |

breakdowns across data loading, forward and backward passes, memory allocation, and inference latency. On ETTh1 and ETTm1, forward and backward times drop by up to an order of magnitude for heavy models such as PDF, MICN, ETSformer, and PPDformer, while DataLoader time remains essentially unchanged. Peak and reserved GPU memory also shrink sharply: on ETTh1 and ETTm1, peak reserved memory decreases from roughly 1.5–2.5 GB to under 0.6 GB for PDF, MICN, ETSformer, and LiNo, and from about 2.6 GB to nearly 1.1 GB for PPDformer. Inference latency shows similar gains, often improving by a factor of 2–5 (e.g., PDF from 0.044 s to 0.022 s on ETTh1 and from 0.049 s to 0.025 s on ETTm1).

Tables 11, 13, 15, 17, and 19 in Appendix A.4 report the corresponding efficiency results for ETTh2, ETTm2, Exchange, Weather, and National Illness, and show the same overall pattern. This indicates that embedding layers are a considerable source of computational overhead in both modern and traditional forecasting models, and that strategically removing them offers a simple way to improve training and inference efficiency without sacrificing accuracy.

**Performance gains increase with horizon length.** The effect of removing the embedding layer increases as the forecasting horizon increases across all four benchmarks. While short-term configurations (e.g., $H = 96$) show limited changes, longer horizons often yield substantial improvements. For instance, on ETTm1, Crossformer shows an MSE reduction of 0.003 at $H = 96$, which increases significantly to 0.365 at $H = 720$. This finding indicates that embedding-free designs may be advantageous for long-term forecasting tasks.

**Architectural sensitivity to embedding layers.** Embedding removal impacts architectural families differently. Transformer-based models rely on self-attention mechanisms to capture long-range dependencies, but they do not include any inherent structure to model sequential order. Unlike recurrent architectures, Transformers require explicit positional and token embeddings to encode temporal progression. In theory, these embeddings are intended to compensate for the lack of built-in sequence modeling. However, our empirical results reveal that removing these embeddings often leads to better performance. MLP-based architectures employ purely feedforward pathways and rely on dense transformations to model dependencies

Table 4: ETTm1 forecasting results with and without embeddings for input length $L = 96$ and prediction horizons $H \in \{96, 192, 336, 720\}$. Bold values indicate better performance.

| Model | Metric | With Embedding | | | | | | Without Embedding | | | | | |
| --- | --- | --- | --- | --- | --- | --- | --- | --- | --- | --- | --- | --- | --- |
| | | H | | | | Time | Mem | H | | | | Time | Mem |
| | | 96 | 192 | 336 | 720 | | | 96 | 192 | 336 | 720 | | |
| PDF | MSE | 0.335 | 0.377 | 0.408 | 0.457 | 194.3 | 2961 | **0.321** | **0.365** | **0.392** | **0.451** | **64.44** | **2880** |
| | MAE | 0.367 | 0.393 | 0.415 | 0.448 | | | **0.359** | **0.386** | **0.405** | **0.442** | | |
| ETSformer | MSE | 0.526 | 0.577 | 0.677 | 0.802 | 94.28 | 2803 | **0.373** | **0.408** | **0.441** | **0.499** | **33.18** | **2200** |
| | MAE | 0.515 | 0.553 | 0.620 | 0.708 | | | **0.397** | **0.410** | **0.429** | **0.462** | | |
| PatchTST | MSE | **0.344** | 0.375 | 0.407 | 0.473 | 58.62 | **2476** | 0.348 | **0.370** | **0.393** | **0.459** | **23.48** | 2893 |
| | MAE | **0.367** | 0.395 | 0.415 | 0.453 | | | 0.371 | **0.387** | **0.406** | **0.440** | | |
| MICN | MSE | **0.320** | 0.378 | 0.428 | 0.483 | 70.17 | 2850 | 0.354 | **0.363** | **0.416** | **0.478** | **17.05** | **2805** |
| | MAE | **0.374** | 0.414 | 0.452 | 0.482 | | | 0.380 | **0.395** | **0.416** | **0.455** | | |
| SOFTS | MSE | 0.325 | 0.384 | 0.429 | 0.477 | 33.51 | **2403** | **0.323** | **0.367** | **0.407** | **0.475** | 27.16 | 2419 |
| | MAE | 0.361 | 0.397 | 0.423 | 0.455 | | | **0.341** | **0.386** | **0.411** | **0.452** | | |
| VarDrop | MSE | 0.340 | 0.398 | 0.439 | **0.490** | 36.26 | 504 | 0.344 | **0.382** | **0.428** | 0.505 | 27.86 | **393** |
| | MAE | 0.375 | 0.403 | 0.427 | **0.457** | | | 0.378 | **0.397** | **0.425** | 0.467 | | |
| Crossformer | MSE | 0.366 | 0.413 | 0.453 | 0.867 | 233.8 | 2237 | **0.363** | **0.406** | **0.447** | **0.511** | 230.4 | **2228** |
| | MAE | 0.406 | 0.427 | 0.454 | 0.711 | | | **0.404** | **0.418** | **0.446** | **0.481** | | |
| iFlashAttention | MSE | 0.350 | 0.402 | 0.442 | 0.500 | 45.11 | 2438 | **0.344** | **0.382** | **0.434** | 0.527 | **34.28** | **2421** |
| | MAE | 0.381 | 0.405 | 0.428 | 0.464 | | | **0.378** | **0.397** | **0.428** | 0.476 | | |
| iFlowformer | MSE | 0.340 | 0.418 | **0.420** | **0.492** | 45.66 | 2409 | **0.339** | **0.388** | 0.448 | 0.501 | **34.97** | **2331** |
| | MAE | 0.373 | 0.412 | **0.424** | **0.461** | | | **0.372** | **0.399** | 0.434 | 0.470 | | |
| PPDformer | MSE | 0.356 | 0.411 | 0.440 | 0.503 | 149.5 | 2840 | **0.339** | **0.389** | **0.432** | **0.493** | 86.75 | 2833 |
| | MAE | 0.392 | 0.420 | 0.438 | 0.471 | | | **0.376** | **0.400** | **0.429** | **0.467** | | |
| LiNo | MSE | 0.331 | 0.400 | 0.435 | 0.503 | 13.50 | 2146 | **0.323** | **0.375** | **0.418** | **0.497** | 12.47 | **2145** |
| | MAE | 0.365 | 0.404 | 0.423 | 0.463 | | | **0.361** | **0.390** | **0.416** | **0.462** | | |
| EDformer | MSE | 0.395 | 0.432 | 0.486 | **0.544** | **10.87** | 2404 | **0.378** | **0.426** | **0.463** | 0.551 | 11.36 | 2805 |
| | MAE | 0.427 | 0.448 | 0.478 | **0.509** | | | **0.413** | **0.443** | **0.466** | 0.520 | | |
| Minusformer | MSE | 0.351 | 0.384 | **0.451** | 0.491 | 40.98 | **2528** | **0.330** | **0.381** | 0.449 | **0.490** | 32.57 | 2847 |
| | MAE | 0.377 | 0.394 | 0.432 | 0.459 | | | **0.367** | **0.392** | **0.428** | **0.457** | | |
| WITRAN | MSE | 0.640 | 0.769 | 0.827 | 0.951 | 107.8 | 937 | **0.637** | **0.688** | **0.803** | **0.860** | 107.6 | **914** |
| | MAE | 0.590 | 0.668 | 0.714 | 0.766 | | | **0.585** | **0.622** | **0.699** | **0.713** | | |
| Times2D | MSE | 0.325 | 0.370 | 0.402 | 0.459 | 21.30 | 783 | **0.324** | **0.368** | **0.397** | **0.458** | 21.09 | **761** |
| | MAE | 0.363 | 0.386 | 0.406 | 0.439 | | | **0.361** | **0.383** | **0.403** | **0.438** | | |
| BiLSTM | MSE | 0.947 | 0.967 | **0.999** | 1.081 | 54.11 | 1703 | **0.928** | **0.964** | 1.005 | **1.056** | **52.59** | **1668** |
| | MAE | 0.686 | **0.7** | **0.726** | 0.782 | | | **0.679** | 0.705 | 0.737 | **0.77** | | |
| ConvLSTM | MSE | 0.937 | 0.988 | 1.02 | 1.08 | **38.85** | 677 | **0.918** | **0.936** | **0.98** | **1.036** | 122.3 | **657** |
| | MAE | 0.686 | 0.72 | 0.749 | 0.789 | | | **0.686** | **0.7** | **0.735** | **0.775** | | |
| GRU | MSE | **0.837** | **0.819** | 1.005 | 1.089 | 35.7 | 665 | 0.931 | 0.961 | **0.989** | **1.07** | 35.27 | **657** |
| | MAE | **0.631** | **0.638** | 0.729 | 0.787 | | | 0.678 | 0.702 | **0.725** | **0.777** | | |
| LSTM | MSE | **0.927** | **0.96** | **1.007** | 1.09 | 39.33 | 668 | 0.968 | 0.99 | 1.02 | **1.072** | **37.8** | 665 |
| | MAE | **0.688** | **0.712** | **0.75** | 0.803 | | | 0.724 | 0.738 | 0.759 | **0.791** | | |
| RNN | MSE | 1.06 | **0.905** | 1.096 | **1.024** | 15.64 | 635 | **0.964** | 1.006 | **1.028** | 1.094 | **15.3** | **632** |
| | MAE | 0.744 | **0.662** | 0.77 | **0.739** | | | **0.703** | 0.733 | **0.755** | 0.797 | | |

Table 5: Average efficiency results for the ETTm1 dataset with and without embeddings, including DataLoader time, forward pass time, backward pass with optimization time, peak allocated GPU memory, peak reserved GPU memory, and inference latency.

| Model | With Embedding | | | | | | Without Embedding | | | | | |
|---|---|---|---|---|---|---|---|---|---|---|---|---|
| | DL | FW | BW | PA | PR | Lat | DL | FW | BW | PA | PR | Lat |
| PDF | 0.064 | 0.116 | 0.18 | 1304 | 1493 | 0.049 | **0.053** | **0.084** | **0.122** | **497.4** | **585.5** | **0.025** |
| MICN | 0.017 | 0.022 | 0.07 | 1266 | 1811 | 0.019 | **0.008** | **0.005** | **0.006** | **102.6** | **117** | **0.004** |
| ETSformer | 0.001 | 0.027 | 0.04 | 2157 | 2456 | 0.033 | **0.001** | **0.008** | **0.012** | **171.7** | **192.5** | **0.006** |
| PatchTST | 0.011 | 0.009 | 0.02 | 445.3 | 545.5 | 0.008 | **0.008** | **0.005** | **0.011** | **184.3** | **190.5** | **0.003** |
| SOFTS | 0.008 | **0.004** | 0.009 | 205.6 | **229** | 0.003 | **0.007** | 0.004 | **0.008** | **164.8** | 319.5 | **0.002** |
| VarDrop | 0.008 | 0.005 | 0.013 | 228.5 | **274** | 0.004 | **0.007** | **0.004** | **0.011** | **166.9** | 322 | **0.003** |
| Crossformer | 0.008 | 0.025 | 0.07 | 1643 | 1909 | 0.017 | **0.007** | **0.024** | **0.065** | **1514** | **1762** | **0.016** |
| FlashAttention | 0.01 | 0.007 | 0.016 | 232.6 | **279.5** | 0.006 | **0.009** | **0.006** | **0.014** | **167.7** | 324 | **0.005** |
| iFlowformer | 0.004 | 0.005 | 0.013 | 259 | **291.5** | 0.004 | **0.004** | **0.005** | **0.012** | **170.4** | 325 | **0.004** |
| PPDformer | 0.034 | 0.058 | 0.074 | 1120 | 2596 | 0.059 | **0.029** | **0.047** | **0.037** | **444** | **1066** | **0.049** |
| LiNo | 0.018 | 0.009 | 0.018 | 210.4 | 251.5 | 0.043 | **0.017** | **0.007** | **0.008** | **58.44** | **71** | **0.04** |
| EDformer | 0.022 | 0.005 | **0.01** | **143.7** | **184** | 0.004 | **0.021** | **0.004** | 0.012 | 201.2 | 360.5 | **0.003** |
| Minusformer | **0.006** | 0.005 | 0.011 | 246.6 | 353 | 0.004 | 0.007 | **0.004** | **0.007** | **161.4** | **201** | **0.003** |
| WITRAN | **0.023** | **0.046** | 0.065 | 46.53 | **61** | **0.043** | 0.023 | 0.048 | **0.064** | 46.53 | 61.5 | 0.043 |
| Times2D | 0.025 | 0.025 | 0.042 | 286.6 | 472.5 | 0.018 | **0.025** | **0.024** | **0.041** | **240.1** | **431.5** | **0.017** |
| BiLSTM | 0.011 | 0.012 | 0.023 | 320.6 | 397 | 0.012 | **0.011** | **0.012** | **0.022** | **300.3** | **369** | **0.012** |
| ConvLSTM | 0.024 | **0.007** | **0.014** | 149.6 | 198 | 0.007 | **0.009** | 0.016 | 0.027 | **143.4** | **184.5** | **0.006** |
| GRU | 0.009 | 0.007 | 0.013 | 143.3 | 181.5 | 0.006 | **0.009** | **0.006** | **0.013** | **132.1** | **174.5** | **0.006** |
| LSTM | 0.008 | 0.007 | 0.014 | 144.8 | 184.5 | 0.007 | **0.008** | **0.007** | **0.014** | **132** | **179.5** | **0.006** |
| RNN | 0.006 | 0.002 | 0.004 | 105.3 | 135.5 | 0.001 | **0.006** | **0.001** | **0.003** | **94.76** | **132.5** | **0.001** |

across time and variables. Since MLPs do not explicitly model sequence order, embedding layers might be expected to play a more important role. Yet, our results show that embeddings are often redundant in MLPs. Hybrid and decomposition-based models incorporate preprocessing such as seasonal-trend decomposition, filtering, or statistical projections. These models are less sensitive to the presence of embedding layers. For example, PDF shows a modest gain. Since these architectures already extract and isolate key patterns before learning begins, embedding layers often duplicate or disrupt this structure, resulting in minimal or inconsistent effects.

**Confidence intervals.** Since deep learning models are inherently stochastic and sensitive to random initialization, we compute 95% confidence intervals (CIs) to assess the statistical reliability of our findings on ETTh1 and ETTm1. The results demonstrate that in all cases—except for LiNo on ETTm1 with $H = 720$ in both MAE and MSE—removing the embedding layers improves performance. Additionally, the corresponding confidence intervals for the models with and without embedding layers do not overlap, indicating statistically significant improvements. Tables 20 and 21 in Appendix A.5 report the MSE and MAE, respectively, along with corresponding 95% confidence intervals for selected high-performing models.

**Configurations with degraded performance.** While the majority of models benefit from removing embedding layers, a few configurations exhibit performance degradation. This outcome can be attributed to several architectural and/or hardware-related factors. First, in the absence of the embedding layer, the model manually permutes, concatenates, and processes the raw input data to reconcile it with the model expected dimensions. These operations introduce additional intermediate tensors and temporary memory allocations, which increase the average memory usage during training. Second, the lower dimensionality resulting from the removal of embedding layers does not align well with the tile sizes optimized in GPU libraries such as cuBLAS, leading to less efficient matrix multiplications and increased computational time. In particular, EDformer originally uses an inverted embedding that transforms the sequence length (e.g., 96) into a typically higher dimension (e.g., 512). EDformer trains approximately 0.3 to 0.5 seconds slower per epoch and consumes an additional 400 MB of memory on average when the embedding layers are removed. The extra permutation,

concatenation, and duplication required to adapt the raw inputs to the expected format increases memory usage. Furthermore, the encoder operates on input tensors with a sequence dimension of 96 instead of 512, which reduces computational throughput due to suboptimal memory access patterns and kernel launch configurations in GPU backends.

## 6 Conclusion

In this paper, we presented a large-scale study assessing the effectiveness of embedding layers in modern time series forecasting models. Our results show that, despite their widespread use, removing data embedding layers from many state-of-the-art forecasting models does not degrade forecasting performance—in many cases, it enhances both forecasting accuracy and computational efficiency. These findings suggest that raw multivariate inputs are often sufficiently informative without the need for additional embedding transformations. Our goal is not to imply that data embedding will never be effective in time series forecasting. Instead, we aim to highlight our promising findings and suggest that the community devote greater attention to critically assessing the actual impact of embedding layers in existing models. For future studies, the effectiveness of embedding layers can be explored on other tasks (e.g., classification, clustering, and imputation) and datasets. Additionally, the effectiveness of other overlooked architectural components—such as normalization strategies, including RevIN—can be investigated.

### Limitations

Here, we outline the limitations of our study:

- We evaluate the impact of data embedding layers specifically for time-series forecasting. However, embedding layers may play different roles in other downstream tasks such as classification, clustering, or imputation, which are not explored in this work.

- The analysis focuses solely on the effect of embedding layers and does not account for potential interactions with other architectural components such as normalization strategies or residual connections.

### Broader impact statement

This work makes a fundamental contribution to time series analysis, particularly in the context of forecasting. It encourages researchers to move beyond default assumptions and critically assess whether each architectural component, such as data embedding layers, meaningfully contributes to performance. Our findings promote a shift in focus: rather than continually developing more complex models, researchers across domains are encouraged to revisit and analyze existing architectures. This approach can lead to significant savings in time, resources, and energy. While our results are limited to forecasting tasks on regularly sampled datasets, the broader methodology—systematic ablation testing of architectural components—can inspire more rigorous empirical validation in other areas of machine learning. We hope this work supports the community in understanding the role and effectiveness of foundational model elements before advancing to further architectural complexity and innovation.

### Author Contributions

Reza Nematirad: Conceptualization, software, investigation, formal analysis, writing - original draft, writing - review & editing. Anil Pahwa: Supervision, project administration, review & editing. Balasubramaniam Natarajan: Supervision, review & editing.

### Acknowledgments

This research was conducted with funding from the NSF under Award No. 2225341.

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

# A  Appendix

## A.1  Embedding layer categories

This section provides a summary of the embedding categories and techniques used in our study.

Table 6: Summary of embedding layer categories and techniques used in this study.

| Category | Technique | Description |
|---|---|---|
| Temporal | Fixed embedding | Maps discrete temporal indices to non-trainable, sinusoidal embeddings. |
| | Learnable embedding | Maps discrete temporal indices (e.g., hour, day) to trainable embeddings. |
| | TimeFeature | Projects numeric time features into high-dimensional space via linear projection. |
| Value | Token embedding | Projects multivariate features to higher dimensions using 1D convolution. |
| | Linear projection | Maps input features directly to embedding space using linear layers. |
| Positional | Sinusoidal positional | Encodes sequence positions using fixed sine and cosine functions. |
| | Learnable positional | Learns embeddings for positional indices in sequences. |
| Combined | Inverted | Fuses variables and time features using linear transformations. |
| Patching | Patchwise encoding | Divides input sequences into patches, encodes each patch, and adds positional info. |

## A.2 Model Invariances When Removing Embeddings

This section clarifies which architectural components change when the embedding layer is removed and which remain fixed. We use $d_{\text{model}}$ to denote the embedding dimension used in the original models, $c_{\text{in}}$ to denote the raw input dimension (number of input features), and seq_len to denote the sequence length (number of time steps). Table 7 summarizes the effects of embedding techniques that transform the feature dimension from $c_{\text{in}}$ to $d_{\text{model}}$, and Table 8 summarizes the effects for embedding techniques that transform the temporal dimension from seq_len to $d_{\text{model}}$.

Table 7: Summary of components affected by the embedding layers that transform the feature dimension from $c_{\text{in}}$ to $d_{\text{model}}$.

| Component | Affected by Removing Embedding? | Notes / Conditions |
|---|---|---|
| Multi-head attention | Yes, projection layers switch from $d_{\text{model}}$ to $c_{\text{in}}$ | Must satisfy $c_{\text{in}} \bmod n_{\text{heads}} = 0$ |
| Feed-forward layers (MLPs) | Yes, input/output dimensions change to $c_{\text{in}}$ | Hidden dimension $d_{\text{ff}}$ stays fixed |
| LayerNorm / residual connections | Yes, normalization width becomes $c_{\text{in}}$ | No change to computation |
| Convolution layers | Yes, `in_channels` becomes $c_{\text{in}}$ | Kernel sizes, strides, and receptive fields unchanged |
| Positional embeddings | Yes, dimension becomes $c_{\text{in}}$ or is disabled | Temporal positions unaffected |
| Number of layers (depth) | No | Encoder/decoder depth unchanged |
| Temporal dimensions | No | seq_len, pred_len, patch length, stride unchanged |
| Attention mechanism type | No | Full/local/flash attention unchanged |
| Receptive field | No | Determined by architecture, not $d_{\text{model}}$ |
| Parameter sharing / weight tying | No | Rules unchanged |

Table 8: Summary of components affected by the embedding layers that transform the temporal dimension from seq_len to $d_{\text{model}}$.

| Component | Affected by Removing Inverted Embedding? | Notes / Conditions |
|---|---|---|
| Multi-head attention | Yes, projection layers switch from $d_{\text{model}}$ to seq_len | Must satisfy seq_len $\bmod \, n_{\text{heads}} = 0$ |
| Feed-forward layers (MLPs) | Yes, input/output dimensions change to seq_len | Hidden dimension $d_{\text{ff}}$ stays fixed |
| LayerNorm / residual connections | Yes, normalization width becomes seq_len | No change to computation |
| Convolution layers | Yes, in_channels becomes seq_len | Kernel sizes, strides, and receptive fields unchanged |
| Temporal encoding | Yes, method changes from learned to concatenated | With: learned temporal embeddings; Without: raw time features |
| Number of layers (depth) | No | Encoder/decoder depth unchanged |
| Channel/variate dimensions | No | $c_{\text{in}} + n_{\text{time\_features}}$ unchanged |
| Attention mechanism type | No | Full/local/flash attention unchanged |
| Sequence dimension for attention | No | Attention operates over $c_{\text{in}} + n_{\text{time\_features}}$ |
| Receptive field | No | Determined by architecture, not $d_{\text{model}}$ |
| Parameter sharing / weight tying | No | Rules unchanged |

### A.3  Benchmark datasets

The benchmark datasets used in this paper cover diverse domains, sampling frequencies, and temporal behaviors, enabling a broad and rigorous evaluation of time series forecasting performance. The Electricity Transformer Temperature (ETT) datasets—ETTh1 and ETTh2 at hourly resolution, and ETTm1 and ETTm2 at 15-minute resolution—contain two years of transformer oil temperature and electrical load-related measurements collected from two counties in China. Each timestamp includes six operational features along with the target oil temperature, providing multivariate sequences.

The weather dataset consists of one year of meteorological observations recorded at 21 weather stations across Germany. It includes 21 meteorological variables sampled every 10 minutes, representing a high-frequency multivariate dataset with strong short-term fluctuations.

The exchange rate dataset includes more than three decades of daily exchange rates for eight major foreign currencies against the U.S. dollar. This dataset captures the volatility and non-stationarity typical of financial time series and provides a long-span, low-frequency benchmark for testing forecasting robustness.

Finally, the National Illness dataset provides weekly influenza-like illness case rates with severe complications collected across U.S. regions from 2002 to 2020. Together, these datasets span high-frequency (10-minute), medium-frequency (15-minute and hourly), daily, and weekly sampling regimes, with diverse feature dimensions and temporal dynamics. This diversity provides a robust foundation for evaluating the impact of data embedding layers on time series forecasting algorithms (Jin et al., 2024).

Table 9: Summary of benchmark datasets used in this study.

| Dataset | Dimension | Train / Val / Test | Frequency | Duration |
|---|---|---|---|---|
| Weather | 21 | (36,600 / 5,079 / 10,348) | 10 minutes | Jan 2020 – Jan 2021 |
| ETTm1 | 7 | (34,465 / 11,521 / 11,521) | 15 minutes | Jul 2016 – Jul 2018 |
| ETTm2 | 7 | (34,465 / 11,521 / 11,521) | 15 minutes | Jul 2016 – Jul 2018 |
| ETTh1 | 7 | (8,545 / 2,881 / 2,881) | 1 hour | Jul 2016 – Jul 2018 |
| ETTh2 | 7 | (8,545 / 2,881 / 2,881) | 1 hour | Jul 2016 – Jul 2018 |
| Exchange Rate | 8 | (40,960 / 5,320 / 11,376) | Daily | Jan 1990 – Oct 2021 |
| National Illness | 7 | (4,067 / 434 / 1,106) | Weekly | Jan 2002 – Jul 2020 |

## A.4 Forecasting results on ETTh2 and ETTm2, Exchange Rate, Weather, and National Illness

This section presents additional forecasting results on the ETTh2, ETTm2, Exchange Rate, Weather, and National Illness datasets, complementing the ETTh1 and ETTm1 results reported in the main text (Table 2 and Table 4). These datasets span a diverse range of temporal resolutions—from high-frequency 10-minute and 15-minute observations to low-frequency weekly and daily records—and exhibit substantially different temporal behaviors, noise patterns, and seasonal structures. Evaluating models on this broader collection of benchmarks allows us to verify whether the trends observed in the main paper generalize beyond the ETTm1 and ETTh1 datasets. Across these datasets, we report the same set of forecasting accuracy metrics (MSE and MAE) over multiple prediction horizons, along with the corresponding computational efficiency metrics. These additional results allow us to examine whether the effects of removing embedding layers remain consistent under different data scales, sampling rates, and domain characteristics. In particular, the Exchange Rate and Illness datasets represent low-resolution, low-dimensional forecasting tasks, whereas ETTm2 and Weather reflect higher-resolution, multivariate inputs.

Table 10: ETTh2 forecasting results with and without embeddings for input length $L = 96$ and prediction horizons $H \in \{96, 192, 336, 720\}$. Bold values indicate better performance.

| Model | Metric | With Embedding | | | | | | Without Embedding | | | | | |
| | | H | | | | Time | Mem | H | | | | Time | Mem |
| | | 96 | 192 | 336 | 720 | | | 96 | 192 | 336 | 720 | | |
| PDF | MSE | 0.307 | 0.376 | 0.414 | 0.437 | 47.82 | 2838 | **0.300** | **0.375** | **0.412** | **0.431** | **16.45** | **2752** |
| | MAE | 0.353 | 0.401 | 0.426 | 0.452 | | | **0.348** | **0.397** | **0.426** | **0.447** | | |
| ETSformer | MSE | 0.399 | 0.521 | 0.615 | 0.692 | 23.50 | 2721 | **0.345** | **0.436** | **0.487** | **0.494** | **8.80** | **2080** |
| | MAE | 0.435 | 0.505 | 0.569 | 0.616 | | | **0.399** | **0.446** | **0.483** | **0.497** | | |
| PatchTST | MSE | 0.300 | 0.382 | 0.435 | 0.447 | 14.96 | **2351** | **0.297** | **0.375** | **0.413** | **0.418** | **6.46** | 2758 |
| | MAE | **0.350** | 0.404 | 0.441 | 0.463 | | | 0.384 | **0.399** | **0.428** | **0.440** | | |
| MICN | MSE | 0.354 | 0.475 | 0.602 | 0.829 | 17.95 | 2761 | **0.333** | **0.463** | **0.567** | **0.804** | **5.24** | **2708** |
| | MAE | 0.400 | 0.477 | 0.540 | 0.654 | | | **0.389** | **0.472** | **0.527** | **0.646** | | |
| SOFTS | MSE | 0.305 | 0.375 | 0.437 | 0.439 | 10.06 | 634 | **0.295** | **0.374** | **0.416** | **0.434** | **8.09** | **473** |
| | MAE | 0.350 | 0.396 | 0.437 | **0.447** | | | **0.347** | **0.395** | **0.430** | 0.450 | | |
| VarDrop | MSE | 0.306 | 0.393 | 0.423 | 0.436 | 9.64 | 2288 | **0.303** | **0.383** | 0.423 | **0.418** | **8.39** | **2274** |
| | MAE | 0.355 | 0.406 | 0.438 | 0.452 | | | **0.353** | **0.400** | **0.436** | **0.442** | | |
| Crossformer | MSE | 0.588 | 0.978 | 0.996 | 1.161 | 56.51 | 2243 | **0.573** | **0.757** | **0.796** | **0.945** | **56.41** | **2223** |
| | MAE | 0.576 | 0.698 | 0.709 | 0.787 | | | **0.537** | **0.628** | **0.643** | **0.803** | | |
| iFlashAttention | MSE | 0.306 | 0.392 | 0.428 | 0.442 | 11.43 | 2303 | **0.305** | **0.382** | **0.426** | **0.417** | **10.00** | **2294** |
| | MAE | 0.355 | 0.406 | 0.438 | 0.455 | | | **0.353** | **0.400** | **0.430** | **0.440** | | |
| iFlowformer | MSE | 0.308 | 0.389 | 0.431 | 0.440 | 13.02 | 2293 | 0.308 | **0.382** | **0.422** | **0.432** | **10.70** | **2281** |
| | MAE | 0.357 | 0.409 | 0.440 | 0.453 | | | **0.355** | **0.402** | **0.433** | **0.447** | | |
| PPDformer | MSE | 0.321 | 0.405 | 0.439 | 0.461 | 36.94 | 2727 | **0.320** | **0.401** | **0.436** | **0.438** | **22.24** | **2714** |
| | MAE | 0.365 | 0.415 | 0.445 | 0.465 | | | **0.361** | **0.406** | **0.439** | **0.454** | | |
| LiNo | MSE | 0.305 | 0.384 | 0.389 | 0.417 | 3.96 | 2035 | **0.296** | **0.378** | **0.385** | **0.412** | **3.83** | **2021** |
| | MAE | 0.352 | 0.398 | 0.413 | 0.436 | | | **0.345** | **0.395** | **0.411** | **0.432** | | |
| EDformer | MSE | 0.422 | 0.485 | 0.549 | 0.799 | **3.66** | **2288** | **0.404** | **0.484** | **0.504** | **0.688** | 3.98 | 2661 |
| | MAE | 0.429 | 0.464 | 0.500 | 0.621 | | | **0.423** | **0.482** | **0.490** | **0.593** | | |
| Minusformer | MSE | 0.304 | 0.379 | 0.430 | 0.427 | 10.94 | 2420 | **0.293** | **0.373** | **0.421** | **0.422** | **8.78** | **2699** |
| | MAE | 0.349 | 0.396 | 0.435 | 0.441 | | | **0.344** | **0.393** | **0.429** | **0.439** | | |
| WITRAN | MSE | 1.659 | 2.810 | 2.641 | 3.488 | 27.12 | 804 | 1.771 | 2.752 | 2.632 | 3.752 | 25.12 | 803 |
| | MAE | 1.055 | 1.464 | 1.415 | 1.663 | | | 1.120 | 1.439 | 1.403 | 1.682 | | |
| Times2D | MSE | **0.292** | 0.376 | 0.379 | 0.413 | 5.52 | 775 | 0.294 | **0.371** | **0.376** | **0.406** | **5.19** | **740** |
| | MAE | **0.340** | 0.391 | 0.407 | 0.434 | | | 0.342 | **0.390** | **0.405** | **0.429** | | |
| BiLSTM | MSE | 1.433 | **1.925** | **2.258** | 2.466 | 13.98 | 1700 | **0.863** | 3.155 | 2.95 | **2.377** | **13.83** | **1661** |
| | MAE | 0.996 | **1.154** | **1.279** | 1.365 | | | **0.74** | 1.478 | 1.478 | **1.23** | | |
| ConvLSTM | MSE | 1.362 | **1.834** | **2.04** | **2.176** | 10.36 | 672 | **0.978** | 2.62 | 2.435 | 2.624 | **10.26** | **661** |
| | MAE | 0.935 | **1.074** | **1.148** | **1.208** | | | **0.784** | 1.354 | 1.29 | 1.34 | | |
| GRU | MSE | 0.947 | **1.717** | **1.926** | 3.377 | 10.26 | 663 | **0.924** | 1.816 | 2.263 | **2.2** | **9.701** | **648** |
| | MAE | 0.782 | **1.041** | **1.112** | 1.615 | | | **0.769** | 1.107 | 1.249 | **1.217** | | |
| LSTM | MSE | **0.902** | 2.631 | 2.842 | 2.903 | 11.5 | 664 | 0.928 | 2.633 | 3.377 | **2.737** | 10.61 | 659 |
| | MAE | **0.758** | 1.399 | **1.466** | 1.476 | | | 0.773 | **1.381** | 1.604 | **1.274** | | |
| RNN | MSE | 0.994 | 2.555 | **1.643** | 1.942 | 5.24 | **633** | 0.962 | 1.963 | 2.434 | 2.265 | 4.81 | 633 |
| | MAE | 0.834 | 1.271 | **0.968** | **1.049** | | | **0.766** | **1.158** | 1.278 | 1.19 | | |

Table 11: Average efficiency results for the ETTh2 dataset with and without embeddings, including DataLoader time, forward pass time, backward pass with optimization time, peak allocated GPU memory, peak reserved GPU memory, and inference latency.

| Model | With Embedding | | | | | | Without Embedding | | | | | |
|---|---|---|---|---|---|---|---|---|---|---|---|---|
| | DL | FW | BW | PA | PR | Lat | DL | FW | BW | PA | PR | Lat |
| PDF | 0.064 | 0.113 | 0.173 | 1304 | 1494 | 0.053 | **0.053** | **0.082** | **0.121** | **497.4** | **585.5** | **0.027** |
| MICN | 0.02 | 0.022 | 0.07 | 1266 | 1811 | 0.019 | **0.012** | **0.007** | **0.007** | **102.6** | **117** | **0.004** |
| ETSformer | 0.001 | 0.027 | 0.041 | 2157 | 2456 | 0.033 | **0.001** | **0.008** | **0.012** | **171.7** | **192.5** | **0.006** |
| PatchTST | 0.014 | 0.009 | 0.02 | 445.3 | 545.5 | 0.008 | **0.012** | **0.005** | **0.011** | **184.3** | **190.5** | **0.003** |
| SOFTS | 0.011 | 0.004 | 0.009 | 205.6 | **229** | 0.003 | **0.011** | **0.004** | **0.008** | **164.8** | 319.5 | **0.002** |
| VarDrop | 0.011 | 0.005 | 0.013 | 228.5 | **274** | 0.004 | **0.011** | **0.005** | **0.011** | **166.9** | 322 | **0.003** |
| Crossformer | 0.008 | 0.024 | 0.065 | 1514 | **1758** | 0.016 | **0.008** | **0.024** | **0.065** | **1514** | 1762 | **0.016** |
| FlashAttention | 0.013 | 0.007 | 0.017 | 232.6 | **279.5** | 0.006 | **0.012** | **0.007** | **0.014** | **167.7** | 324 | **0.005** |
| iFlowformer | 0.006 | 0.006 | 0.013 | 259 | **291.5** | 0.004 | **0.006** | **0.005** | **0.012** | **170.4** | 325 | **0.004** |
| PPDformer | 0.047 | 0.062 | 0.097 | 1290 | 2888 | 0.058 | **0.041** | **0.051** | **0.042** | **461.6** | **1084** | **0.047** |
| LiNo | 0.031 | 0.01 | 0.018 | 210.4 | 251.5 | 0.043 | **0.029** | **0.007** | **0.009** | **58.44** | **71** | **0.038** |
| EDformer | **0.043** | 0.005 | **0.01** | **143.7** | **184** | 0.004 | 0.045 | **0.004** | 0.012 | 201.2 | 360.5 | **0.003** |
| Minusformer | 0.01 | 0.005 | 0.011 | 246.6 | 353 | 0.004 | **0.008** | **0.004** | **0.007** | **161.4** | **201** | **0.003** |
| WITRAN | 0.027 | **0.048** | **0.064** | 46.53 | **61** | 0.043 | **0.026** | 0.049 | 0.065 | 46.53 | 61.5 | **0.043** |
| Times2D | 0.04 | 0.026 | 0.042 | 286.6 | 472 | 0.018 | **0.04** | **0.025** | **0.04** | **240.1** | **431.5** | **0.018** |
| BiLSTM | 0.014 | 0.012 | 0.023 | 320.6 | 397 | 0.012 | **0.014** | **0.012** | **0.022** | **300.3** | **369** | **0.012** |
| ConvLSTM | **0.013** | 0.008 | 0.052 | 149.6 | 198 | 0.007 | 0.013 | **0.007** | **0.014** | **143.4** | **184.5** | **0.006** |
| GRU | 0.013 | 0.007 | 0.013 | 143.3 | 181.5 | 0.006 | **0.013** | **0.006** | **0.013** | **132.1** | **174.5** | **0.006** |
| LSTM | 0.011 | 0.007 | 0.014 | 144.8 | 184.5 | 0.007 | **0.011** | **0.007** | **0.014** | **132** | **179** | **0.006** |
| RNN | 0.01 | 0.002 | 0.004 | 105.3 | 135.5 | 0.001 | **0.01** | **0.001** | **0.003** | **94.76** | **132.5** | **0.001** |

Table 12: ETTm2 forecasting results with and without embeddings for input length $L = 96$ and prediction horizons $H \in \{96, 192, 336, 720\}$. Bold values indicate better performance.

| Model | Metric | With Embedding | | | | Time | Mem | Without Embedding | | | | Time | Mem |
|---|---|---|---|---|---|---|---|---|---|---|---|---|---|
| | | H | | | | | | H | | | | | |
| | | 96 | 192 | 336 | 720 | | | 96 | 192 | 336 | 720 | | |
| PDF | MSE | 0.183 | 0.246 | 0.300 | 0.403 | 193.9 | 2975 | **0.176** | **0.241** | 0.304 | **0.402** | **66.29** | **2877** |
| | MAE | 0.265 | 0.307 | 0.342 | 0.401 | | | **0.258** | **0.301** | 0.344 | 0.403 | | |
| ETSformer | MSE | 0.267 | 0.333 | 0.398 | 0.501 | 92.92 | 2806 | **0.189** | **0.255** | **0.318** | **0.432** | **32.23** | **2209** |
| | MAE | 0.372 | 0.409 | 0.444 | 0.495 | | | **0.284** | **0.323** | **0.361** | **0.427** | | |
| PatchTST | MSE | 0.183 | 0.248 | **0.311** | **0.407** | 58.71 | 2472 | **0.177** | **0.244** | 0.314 | 0.410 | **24.58** | **2458** |
| | MAE | 0.263 | 0.309 | 0.351 | **0.402** | | | **0.261** | **0.305** | **0.350** | 0.408 | | |
| MICN | MSE | **0.183** | **0.272** | 0.396 | 0.579 | 70.61 | 2845 | 0.193 | 0.280 | **0.314** | **0.508** | **17.24** | **2836** |
| | MAE | **0.280** | **0.346** | 0.429 | 0.532 | | | 0.292 | 0.358 | **0.350** | **0.498** | | |
| SOFTS | MSE | 0.180 | 0.251 | 0.315 | 0.417 | 30.73 | 638 | **0.179** | **0.247** | **0.309** | **0.411** | **28.43** | **481** |
| | MAE | 0.262 | 0.309 | 0.349 | 0.407 | | | **0.263** | **0.308** | **0.346** | **0.405** | | |
| VarDrop | MSE | 0.182 | 0.250 | **0.312** | **0.410** | 36.48 | 499 | **0.181** | **0.250** | 0.322 | 0.421 | **28.34** | **391** |
| | MAE | 0.266 | 0.311 | **0.350** | **0.405** | | | **0.265** | **0.309** | 0.357 | 0.410 | | |
| Crossformer | MSE | **0.237** | 0.450 | **0.640** | 1.660 | 232.8 | 2145 | 0.257 | **0.437** | 0.701 | **1.520** | **232.7** | **2131** |
| | MAE | 0.342 | 0.462 | **0.548** | 0.914 | | | **0.336** | **0.442** | 0.602 | **0.886** | | |
| iFlashAttention | MSE | **0.182** | 0.250 | **0.312** | 0.411 | 39.93 | 2389 | 0.194 | **0.249** | 0.322 | 0.416 | **33.69** | **2375** |
| | MAE | **0.266** | 0.311 | **0.349** | **0.405** | | | 0.279 | **0.309** | 0.357 | 0.408 | | |
| iFlowformer | MSE | 0.183 | 0.249 | 0.311 | **0.409** | 32.70 | 2391 | **0.181** | **0.248** | **0.312** | 0.420 | **27.12** | **2391** |
| | MAE | 0.269 | 0.310 | 0.349 | **0.404** | | | **0.267** | **0.307** | **0.348** | 0.410 | | |
| PPDformer | MSE | 0.188 | 0.269 | 0.322 | 0.417 | 148.7 | 2870 | **0.180** | **0.254** | **0.308** | **0.407** | **86.57** | **2802** |
| | MAE | 0.276 | 0.329 | 0.357 | 0.411 | | | **0.260** | **0.307** | **0.345** | **0.403** | | |
| LiNo | MSE | 0.177 | 0.244 | 0.309 | 0.404 | 13.98 | **2147** | **0.173** | **0.241** | **0.304** | **0.403** | **11.19** | 2171 |
| | MAE | 0.260 | 0.304 | 0.346 | **0.398** | | | **0.256** | **0.301** | **0.342** | 0.399 | | |
| EDformer | MSE | 0.310 | 0.500 | **0.647** | **0.755** | 12.34 | **2397** | 0.262 | 0.449 | 0.668 | 0.776 | **10.93** | 2804 |
| | MAE | 0.388 | 0.492 | **0.590** | 0.637 | | | **0.353** | **0.474** | 0.607 | **0.617** | | |
| Minusformer | MSE | 0.183 | 0.248 | 0.309 | 0.409 | 41.26 | 2543 | **0.176** | **0.246** | **0.308** | **0.401** | **33.69** | **2848** |
| | MAE | 0.268 | 0.308 | 0.347 | 0.402 | | | **0.260** | **0.304** | **0.345** | **0.400** | | |
| WITRAN | MSE | 0.807 | 1.136 | **1.293** | 4.448 | 109.5 | 943 | **0.795** | **1.092** | 1.313 | **4.439** | 107.5 | **925** |
| | MAE | 0.722 | 0.903 | **0.916** | 1.793 | | | **0.709** | **0.886** | 0.965 | **1.628** | | |
| Times2D | MSE | 0.179 | 0.241 | 0.301 | 0.397 | **20.72** | 781 | **0.175** | **0.240** | **0.300** | **0.394** | 21.53 | **764** |
| | MAE | 0.263 | 0.301 | 0.339 | 0.394 | | | **0.256** | **0.299** | **0.338** | **0.392** | | |
| BiLSTM | MSE | 0.419 | 0.599 | **0.895** | 2.205 | 54.71 | 1705 | **0.346** | **0.574** | 0.902 | **1.641** | **52.75** | **1662** |
| | MAE | 0.477 | 0.602 | 0.762 | 1.23 | | | **0.442** | **0.591** | **0.753** | **1.045** | | |
| ConvLSTM | MSE | **0.395** | **0.6** | 1.303 | **1.855** | 91.44 | 685 | 0.549 | 0.765 | **1.082** | 2.208 | **36.62** | **670** |
| | MAE | **0.465** | **0.617** | 0.908 | **1.109** | | | 0.589 | 0.712 | **0.858** | 1.259 | | |
| GRU | MSE | **0.321** | 0.698 | **0.883** | **1.478** | 36.14 | 667 | 0.375 | **0.647** | 1.035 | 1.52 | **35.36** | **656** |
| | MAE | **0.415** | **0.646** | **0.745** | **0.968** | | | 0.469 | 0.65 | 0.795 | 1.001 | | |
| LSTM | MSE | **0.338** | 0.519 | **0.846** | 1.695 | 39.92 | 670 | 0.418 | **0.503** | 0.879 | **1.685** | **36.15** | **662** |
| | MAE | **0.428** | **0.548** | **0.735** | 1.076 | | | 0.495 | 0.551 | 0.737 | **1.067** | | |
| RNN | MSE | **0.456** | **0.578** | 1.033 | 2.368 | 16.53 | **634** | 0.617 | 0.734 | **0.869** | **1.294** | 14.51 | 636 |
| | MAE | **0.498** | **0.58** | 0.809 | 1.223 | | | 0.607 | 0.668 | **0.743** | **0.913** | | |

Table 13: Average efficiency results for the ETTm2 dataset with and without embeddings, including DataLoader time, forward pass time, backward pass with optimization time, peak allocated GPU memory, peak reserved GPU memory, and inference latency.

| Model | With Embedding | | | | | | Without Embedding | | | | | |
|---|---|---|---|---|---|---|---|---|---|---|---|---|
| | DL | FW | BW | PA | PR | Lat | DL | FW | BW | PA | PR | Lat |
| PDF | 0.065 | 0.116 | 0.182 | 1304 | 1494 | 0.048 | **0.053** | **0.082** | **0.122** | **497.4** | **585.5** | **0.024** |
| MICN | 0.017 | 0.022 | 0.07 | 1266 | 1811 | 0.019 | **0.008** | **0.005** | **0.006** | **102.6** | **117** | **0.005** |
| ETSformer | 0.001 | 0.027 | 0.04 | 2157 | 2456 | 0.033 | **0.001** | **0.008** | **0.012** | **171.7** | **192.5** | **0.006** |
| PatchTST | 0.011 | 0.009 | 0.02 | 445.3 | 545.5 | 0.008 | **0.008** | **0.005** | **0.011** | **184.3** | **190.5** | **0.003** |
| SOFTS | 0.008 | 0.004 | 0.009 | 205.6 | **229** | 0.003 | **0.007** | **0.003** | **0.008** | **164.8** | 319.5 | **0.002** |
| VarDrop | **0.008** | 0.005 | 0.013 | 228.5 | **274** | 0.004 | 0.008 | **0.004** | **0.011** | **166.9** | 322 | **0.003** |
| Crossformer | **0.007** | **0.024** | 0.065 | 1514 | **1757** | 0.016 | 0.007 | 0.024 | **0.065** | 1514 | 1762 | 0.016 |
| FlashAttention | 0.01 | 0.007 | 0.017 | 232.6 | **279.5** | 0.006 | **0.009** | **0.006** | **0.014** | **167.7** | 324 | **0.005** |
| iFlowformer | 0.01 | 0.008 | 0.019 | 259 | **291.5** | 0.006 | **0.01** | **0.007** | **0.015** | **170.4** | 324.5 | **0.006** |
| PPDformer | 0.023 | 0.06 | 0.08 | 1206 | 2708 | 0.059 | **0.02** | **0.05** | **0.042** | **448.3** | **1080** | **0.049** |
| LiNo | 0.019 | 0.01 | 0.018 | 210.4 | 251 | 0.043 | **0.017** | **0.007** | **0.008** | **58.44** | **71** | **0.04** |
| EDformer | 0.022 | 0.005 | **0.01** | **143.7** | **184** | 0.004 | **0.02** | **0.004** | 0.012 | 201.2 | 360.5 | **0.003** |
| Minusformer | 0.006 | 0.005 | 0.011 | 246.6 | 353 | 0.004 | **0.006** | **0.004** | **0.007** | **161.4** | **201** | **0.003** |
| WITRAN | **0.026** | **0.046** | **0.062** | 46.53 | **61** | **0.041** | 0.027 | 0.046 | 0.063 | 46.53 | 61.5 | 0.042 |
| Times2D | 0.025 | 0.025 | 0.042 | 286.6 | 472 | 0.018 | **0.025** | **0.025** | **0.04** | **240.1** | **431** | **0.017** |
| BiLSTM | 0.011 | 0.012 | 0.023 | 320.6 | 397 | 0.012 | **0.011** | **0.012** | **0.022** | **300.3** | **369** | **0.012** |
| ConvLSTM | 0.009 | 0.007 | 0.02 | 149.6 | 198 | 0.007 | **0.008** | **0.007** | **0.014** | **143.4** | **184.5** | **0.006** |
| GRU | 0.009 | 0.007 | 0.013 | 143.3 | 181.5 | 0.006 | **0.009** | **0.006** | **0.013** | **132.1** | **174.5** | **0.006** |
| LSTM | 0.008 | 0.007 | 0.014 | 144.8 | 184.5 | 0.007 | **0.008** | **0.007** | **0.014** | **132** | **179.5** | **0.006** |
| RNN | **0.006** | 0.002 | 0.004 | 105.3 | 135.5 | 0.001 | 0.014 | **0.001** | **0.003** | **94.76** | **132.5** | **0.001** |

Table 14: Exchange Rate forecasting results with and without embeddings for prediction horizons $H \in \{96, 192, 336, 720\}$. Bold values indicate better performance.

| Model | Metric | With Embedding | | | | | | Without Embedding | | | | | |
|---|---|---|---|---|---|---|---|---|---|---|---|---|---|
| | | H | | | | Time | Mem | H | | | | Time | Mem |
| | | 96 | 192 | 336 | 720 | | | 96 | 192 | 336 | 720 | | |
| PDF | MSE | **0.083** | **0.176** | 0.336 | 0.969 | 64.11 | 545 | 0.085 | 0.178 | **0.332** | **0.895** | 59.31 | **454** |
| | MAE | **0.201** | 0.299 | 0.418 | 0.73 | | | 0.202 | **0.298** | **0.416** | **0.71** | | |
| MICN | MSE | 0.097 | 0.212 | 0.369 | **0.69** | 143.3 | 1496 | **0.079** | **0.155** | **0.292** | 0.759 | 36.1 | 384 |
| | MAE | 0.22 | 0.349 | 0.465 | **0.638** | | | **0.203** | **0.294** | **0.415** | 0.668 | | |
| ETSformer | MSE | 0.129 | 0.217 | 0.369 | 0.892 | 196.6 | 1694 | **0.091** | **0.188** | **0.358** | **0.576** | 62.59 | 367 |
| | MAE | 0.272 | 0.349 | 0.455 | 0.726 | | | **0.216** | **0.323** | **0.452** | **0.598** | | |
| PatchTST | MSE | **0.084** | **0.175** | **0.324** | 1.138 | 67.92 | 734 | 0.084 | 0.181 | 0.327 | **0.913** | 45.23 | 379 |
| | MAE | **0.202** | **0.297** | **0.412** | 0.783 | | | 0.203 | 0.304 | 0.415 | **0.717** | | |
| SOFTS | MSE | **0.086** | **0.18** | **0.329** | 0.919 | 36.31 | **385** | 0.099 | 0.187 | 0.344 | **0.844** | 31.51 | 395 |
| | MAE | **0.206** | **0.302** | **0.415** | 0.719 | | | 0.222 | 0.309 | 0.427 | **0.696** | | |
| VarDrop | MSE | 0.11 | 0.191 | 0.345 | **0.85** | 44.84 | 412 | **0.088** | **0.183** | **0.34** | 0.868 | 37.28 | 391 |
| | MAE | 0.238 | 0.318 | 0.427 | **0.698** | | | **0.209** | **0.305** | **0.423** | 0.704 | | |
| Crossformer | MSE | 0.302 | 0.754 | 1.411 | **1.159** | 386.1 | 1711 | **0.218** | **0.436** | **1.098** | 1.307 | 382.8 | 1685 |
| | MAE | 0.421 | 0.647 | 0.948 | **0.877** | | | **0.341** | **0.506** | **0.801** | 0.901 | | |
| FlashAttention | MSE | 0.112 | 0.184 | **0.326** | 0.88 | 49.59 | 422 | **0.088** | **0.183** | 0.34 | **0.868** | 43.84 | 406 |
| | MAE | 0.24 | 0.311 | **0.416** | 0.709 | | | **0.209** | **0.305** | 0.424 | **0.704** | | |
| iFlowformer | MSE | 0.09 | 0.188 | **0.336** | 0.8 | 58.63 | 448 | **0.088** | **0.183** | 0.337 | 0.867 | 54.22 | 410 |
| | MAE | 0.212 | 0.311 | 0.422 | **0.675** | | | **0.209** | **0.304** | **0.422** | 0.704 | | |
| PPDformer | MSE | **0.103** | **0.203** | 0.371 | 1.022 | 214.2 | 911 | 0.112 | 0.218 | **0.357** | **0.903** | 148.3 | 583 |
| | MAE | **0.228** | **0.321** | 0.442 | 0.764 | | | 0.232 | 0.331 | **0.432** | **0.719** | | |
| LiNo | MSE | 0.087 | 0.186 | 0.346 | 0.928 | 43.48 | 305 | **0.085** | **0.178** | **0.328** | **0.83** | 38.56 | 280 |
| | MAE | 0.206 | 0.308 | 0.427 | 0.733 | | | **0.204** | **0.299** | **0.416** | **0.685** | | |
| EDformer | MSE | **0.098** | 0.296 | **0.752** | **0.909** | 18.63 | 412 | 0.12 | **0.219** | 1.113 | 0.96 | 16.52 | 348 |
| | MAE | **0.23** | 0.382 | **0.611** | **0.737** | | | 0.239 | **0.327** | 0.73 | 0.745 | | |
| Minusformer | MSE | 0.087 | **0.176** | **0.316** | 1.207 | 68.71 | 555 | **0.084** | 0.177 | 0.331 | **0.854** | 59.23 | 437 |
| | MAE | 0.207 | 0.299 | **0.407** | 0.814 | | | **0.204** | **0.298** | 0.416 | **0.696** | | |
| WITRAN | MSE | **0.85** | **0.963** | 1.732 | **3.153** | 179.9 | **292** | 0.894 | 0.985 | **1.698** | 3.183 | 173.1 | 294 |
| | MAE | **0.762** | **0.791** | 1.1 | **1.498** | | | 0.783 | 0.8 | **1.088** | 1.505 | | |
| Times2D | MSE | 0.082 | **0.172** | 0.327 | 0.847 | 42.7 | 479 | **0.082** | 0.175 | **0.325** | **0.837** | 42.08 | 451 |
| | MAE | 0.199 | **0.293** | 0.413 | 0.692 | | | **0.198** | 0.296 | **0.412** | **0.688** | | |
| BiLSTM | MSE | 0.541 | 0.849 | 1.401 | 2.094 | 57.76 | 880 | **0.36** | **0.539** | **0.757** | **0.943** | 55.98 | 843 |
| | MAE | 0.595 | 0.747 | 0.983 | 1.216 | | | **0.486** | **0.602** | **0.732** | **0.815** | | |
| ConvLSTM | MSE | **0.51** | 1.156 | 1.501 | 2.426 | 104.1 | 692 | 0.553 | **0.828** | **1.032** | **1.843** | 42.7 | 674 |
| | MAE | **0.585** | 0.889 | 0.967 | 1.31 | | | 0.611 | **0.758** | **0.835** | **1.135** | | |
| GRU | MSE | 0.465 | 1.234 | 0.921 | 1.187 | 121.6 | 670 | **0.388** | **0.585** | **0.837** | **1.126** | 34.46 | 662 |
| | MAE | 0.563 | 0.918 | 0.809 | 0.932 | | | **0.503** | **0.63** | **0.765** | **0.879** | | |
| LSTM | MSE | **0.589** | 1.099 | 1.168 | 1.778 | 35.21 | 1497 | 0.596 | **0.959** | **0.86** | **1.185** | 34.43 | 1494 |
| | MAE | **0.609** | 0.867 | 0.886 | 1.064 | | | 0.653 | **0.781** | **0.786** | **0.91** | | |
| RNN | MSE | 0.586 | **0.68** | **0.839** | 1.116 | 15.68 | 631 | **0.517** | 1.493 | 0.904 | **1.105** | 14.9 | 630 |
| | MAE | 0.662 | **0.715** | **0.788** | 0.895 | | | **0.59** | 1.057 | 0.803 | **0.877** | | |

Table 15: Average efficiency results for the Exchange Rate dataset with and without embeddings, including DataLoader time, forward pass time, backward pass with optimization time, peak allocated GPU memory, peak reserved GPU memory, and inference latency.

| Model | With Embedding | | | | | | Without Embedding | | | | | |
|---|---|---|---|---|---|---|---|---|---|---|---|---|
| | DL | FW | BW | PA | PR | Lat | DL | FW | BW | PA | PR | Lat |
| PDF | **0.014** | 0.025 | 0.048 | 446.9 | 504 | 0.018 | 0.014 | **0.023** | **0.041** | 302.5 | 334.5 | **0.016** |
| MICN | 0.01 | 0.022 | 0.069 | 1266 | 1807 | 0.018 | **0.008** | **0.004** | **0.008** | 166.8 | 241.5 | **0.003** |
| ETSformer | 0.015 | 0.059 | 0.072 | 2157 | 2455 | 0.057 | **0.009** | **0.012** | **0.017** | 173.1 | 194.5 | **0.009** |
| PatchTST | 0.009 | 0.01 | 0.024 | 472.2 | 591 | 0.008 | **0.007** | **0.005** | **0.013** | **200.6** | **228** | **0.004** |
| SOFTS | 0.007 | 0.004 | 0.009 | 208.7 | **230.5** | 0.003 | **0.007** | **0.004** | **0.008** | 165.6 | 300 | **0.002** |
| VarDrop | **0.007** | 0.005 | 0.013 | 231.2 | **259** | 0.004 | 0.01 | **0.004** | **0.01** | 168.2 | 303 | **0.003** |
| Crossformer | 0.011 | 0.028 | 0.117 | 1640 | 1866 | 0.025 | **0.011** | **0.028** | **0.112** | 1639 | 1863 | 0.025 |
| FlashAttention | 0.008 | 0.006 | 0.016 | 235.6 | **266.5** | 0.006 | **0.007** | **0.006** | **0.013** | 169 | 304.5 | **0.005** |
| iFlowformer | **0.008** | 0.008 | 0.019 | 263.2 | **288.5** | 0.006 | 0.008 | **0.007** | **0.016** | 171.8 | 306 | **0.006** |
| PPDformer | 0.018 | 0.062 | 0.074 | 1104 | 2160 | 0.057 | **0.013** | **0.05** | **0.036** | 419.6 | 953.5 | **0.05** |
| LiNo | 0.008 | 0.009 | 0.008 | 56.93 | 64 | 0.041 | **0.007** | **0.005** | **0.008** | **31.7** | **37** | **0.004** |
| EDformer | **0.017** | 0.005 | **0.011** | **148.6** | **195.5** | 0.004 | 0.018 | **0.005** | 0.013 | 202.5 | 362.5 | **0.003** |
| Minusformer | 0.005 | 0.005 | 0.012 | 246.9 | 352.5 | 0.004 | **0.005** | **0.004** | **0.009** | **167.7** | **336** | **0.004** |
| WITRAN | 0.021 | 0.047 | **0.067** | 46.75 | **61** | **0.044** | **0.017** | **0.047** | 0.067 | 46.75 | 61.5 | 0.044 |
| Times2D | **0.022** | **0.027** | 0.045 | 321.6 | 518 | 0.019 | 0.023 | 0.027 | **0.042** | **267.2** | **459** | **0.018** |
| BiLSTM | 0.013 | 0.012 | 0.023 | 320.8 | 398 | 0.012 | **0.007** | **0.012** | **0.022** | **300.5** | **369.5** | **0.011** |
| ConvLSTM | 0.005 | 0.007 | 0.028 | 149.8 | 198 | 0.007 | **0.005** | **0.007** | **0.021** | **143.6** | **184.5** | **0.006** |
| GRU | 0.012 | 0.007 | 0.025 | 143.6 | 182 | 0.006 | **0.012** | **0.006** | **0.019** | **132.3** | **174.5** | **0.006** |
| LSTM | 0.005 | 0.007 | 0.014 | 145 | 184.5 | 0.007 | **0.005** | **0.006** | **0.013** | **132.2** | **180** | **0.006** |
| RNN | **0.004** | 0.002 | 0.004 | 105.5 | 135.5 | 0.001 | 0.004 | **0.001** | **0.003** | **94.99** | **133** | **0.001** |

Table 16: Weather forecasting results with and without embeddings for prediction horizons $H \in \{96, 192, 336, 720\}$. Bold values indicate better performance.

| Model | Metric | With Embedding H 96 | 192 | 336 | 720 | Time | Mem | Without Embedding H 96 | 192 | 336 | 720 | Time | Mem |
|---|---|---|---|---|---|---|---|---|---|---|---|---|---|
| PDF | MSE | **0.175** | **0.22** | **0.276** | **0.35** | 98.54 | 881 | 0.178 | 0.224 | 0.279 | 0.354 | **89.46** | **778** |
|  | MAE | **0.217** | **0.255** | **0.296** | **0.346** |  |  | 0.219 | 0.258 | 0.298 | 0.347 |  |  |
| MICN | MSE | 0.192 | 0.233 | 0.275 | 0.327 | 40.29 | 776 | **0.186** | **0.226** | **0.261** | **0.31** | **38.76** | **728** |
|  | MAE | 0.265 | 0.3 | 0.333 | 0.371 |  |  | **0.255** | **0.291** | **0.313** | **0.349** |  |  |
| VarDrop | MSE | 0.196 | 0.243 | 0.293 | 0.364 | **49.43** | **423** | **0.178** | **0.224** | **0.284** | **0.358** | 57.7 | 510 |
|  | MAE | 0.234 | 0.273 | 0.308 | 0.355 |  |  | **0.218** | **0.258** | **0.301** | **0.351** |  |  |
| Crossformer | MSE | 0.16 | **0.204** | 0.274 | 0.401 | 327.3 | **2068** | **0.153** | 0.207 | **0.272** | **0.353** | **321.2** | 2082 |
|  | MAE | 0.231 | 0.274 | 0.334 | 0.404 |  |  | **0.224** | **0.271** | **0.323** | **0.379** |  |  |
| FlashAttention | MSE | 0.195 | 0.243 | 0.293 | 0.364 | **56.08** | **456** | **0.176** | **0.222** | **0.283** | **0.356** | 72.95 | 578 |
|  | MAE | 0.234 | 0.273 | 0.308 | 0.355 |  |  | **0.215** | **0.255** | **0.3** | **0.35** |  |  |
| iFlowformer | MSE | 0.173 | 0.227 | **0.279** | 0.359 | 62.12 | **438** | 0.172 | 0.227 | 0.283 | **0.357** | 59.96 | 519 |
|  | MAE | 0.215 | 0.261 | **0.298** | 0.352 |  |  | **0.211** | **0.26** | 0.3 | **0.349** |  |  |
| PPDformer | MSE | 0.195 | 0.239 | 0.293 | 0.362 | 454.7 | 1492 | **0.16** | **0.21** | **0.27** | **0.348** | **268.6** | **797** |
|  | MAE | 0.243 | 0.277 | 0.314 | 0.357 |  |  | **0.204** | **0.254** | **0.297** | **0.348** |  |  |
| LiNo | MSE | 0.163 | **0.205** | **0.262** | 0.349 | 22.12 | 842 | **0.159** | 0.207 | 0.265 | **0.346** | **16.62** | **624** |
|  | MAE | 0.207 | **0.247** | **0.289** | 0.347 |  |  | **0.204** | 0.249 | 0.294 | **0.347** |  |  |
| EDformer | MSE | 0.201 | 0.25 | 0.292 | 0.354 | 56.67 | 554 | **0.172** | **0.21** | **0.263** | **0.331** | **37.66** | **527** |
|  | MAE | 0.268 | 0.317 | 0.348 | 0.396 |  |  | **0.229** | **0.266** | **0.31** | **0.358** |  |  |
| Minusformer | MSE | **0.17** | 0.223 | 0.282 | **0.355** | 83.18 | 562 | 0.175 | **0.223** | **0.279** | 0.356 | **78.74** | **526** |
|  | MAE | **0.21** | **0.257** | 0.3 | **0.348** |  |  | 0.214 | 0.257 | **0.298** | 0.349 |  |  |
| WITRAN | MSE | **0.502** | 0.439 | 0.44 | 0.604 | **184.8** | **291** | 0.506 | **0.395** | **0.398** | **0.55** | 186 | 293 |
|  | MAE | 0.526 | 0.477 | 0.469 | 0.563 |  |  | **0.52** | **0.439** | **0.432** | **0.53** |  |  |
| Times2D | MSE | 0.181 | 0.232 | 0.285 | 0.357 | 187.1 | 6137 | **0.179** | **0.23** | **0.281** | **0.356** | **170** | **5358** |
|  | MAE | 0.233 | 0.262 | 0.3 | 0.347 |  |  | **0.229** | **0.26** | **0.297** | **0.347** |  |  |
| BiLSTM | MSE | 0.239 | **0.272** | 0.336 | 0.418 | 70.51 | 1218 | **0.239** | 0.282 | **0.329** | **0.398** | **69.12** | **1161** |
|  | MAE | **0.328** | **0.35** | 0.398 | 0.453 |  |  | 0.328 | 0.363 | **0.396** | **0.438** |  |  |
| ConvLSTM | MSE | 0.297 | 0.313 | 0.387 | 0.469 | 55.02 | 849 | **0.292** | **0.29** | **0.338** | **0.405** | **52.24** | **843** |
|  | MAE | **0.374** | 0.386 | 0.437 | 0.486 |  |  | 0.376 | **0.371** | **0.404** | **0.445** |  |  |
| GRU | MSE | 0.576 | 0.291 | 0.35 | 0.435 | 45.24 | 807 | **0.195** | **0.256** | **0.301** | **0.415** | **45.16** | **785** |
|  | MAE | 0.563 | 0.367 | 0.405 | 0.459 |  |  | **0.283** | **0.338** | **0.375** | **0.45** |  |  |
| LSTM | MSE | 0.288 | 0.297 | 0.353 | 0.432 | **44.05** | 1668 | **0.232** | **0.278** | **0.315** | **0.413** | 44.24 | **1624** |
|  | MAE | 0.363 | 0.37 | 0.41 | 0.458 |  |  | **0.321** | **0.359** | **0.378** | **0.449** |  |  |
| RNN | MSE | 0.423 | 0.37 | 0.347 | 0.858 | **40** | 683 | **0.228** | **0.29** | **0.341** | **0.434** | 40.26 | **655** |
|  | MAE | 0.477 | 0.436 | 0.407 | 0.723 |  |  | **0.315** | **0.371** | **0.403** | **0.465** |  |  |

Table 17: Average efficiency results for the Weather dataset with and without embeddings, including DataLoader time, forward pass time, backward pass with optimization time, peak allocated GPU memory, peak reserved GPU memory, and inference latency.

| Model | With Embedding | | | | | | Without Embedding | | | | | |
|---|---|---|---|---|---|---|---|---|---|---|---|---|
| | DL | FW | BW | PA | PR | Lat | DL | FW | BW | PA | PR | Lat |
| PDF | 0.01 | 0.017 | 0.036 | 581.4 | 622.5 | 0.011 | **0.01** | **0.017** | **0.031** | **384.4** | **420** | **0.01** |
| MICN | **0.008** | 0.004 | 0.01 | 430.1 | 571 | **0.003** | 0.008 | **0.004** | **0.009** | **307.2** | **509** | 0.003 |
| ETSformer | 0.007 | 0.021 | 0.025 | 635.4 | 756 | 0.027 | **0.006** | **0.008** | **0.011** | **127.1** | **146** | **0.006** |
| PatchTST | **0.003** | 0.003 | 0.005 | **91.53** | **112** | 0.002 | 0.003 | **0.002** | **0.005** | 177.5 | 286.5 | **0.002** |
| SOFTS | 0.005 | 0.002 | **0.004** | **103.7** | **130.5** | 0.002 | **0.005** | **0.002** | 0.007 | 164 | 315.5 | **0.002** |
| VarDrop | **0.011** | 0.006 | 0.012 | **143.8** | **186** | 0.004 | 0.011 | **0.005** | **0.011** | 181.4 | 436 | **0.003** |
| Crossformer | **0.013** | 0.027 | 0.095 | 1342 | 1474 | **0.021** | 0.013 | **0.027** | **0.091** | **1339** | **1472** | 0.021 |
| FlashAttention | **0.011** | **0.007** | **0.014** | **160** | **205** | **0.006** | 0.012 | 0.01 | 0.02 | 185.4 | 441.5 | 0.008 |
| iFlowformer | 0.012 | 0.009 | 0.018 | 196.6 | **245** | 0.007 | **0.011** | **0.007** | **0.017** | **186.2** | 440.5 | **0.006** |
| PPDformer | **0.037** | 0.122 | 0.182 | 2188 | 4088 | 0.111 | 0.039 | **0.09** | **0.068** | **810.6** | **1456** | **0.084** |
| LiNo | 0.02 | 0.013 | 0.023 | 444.5 | 547 | 0.036 | **0.02** | **0.011** | **0.008** | **134.1** | **162.5** | **0.032** |
| EDformer | 0.016 | 0.005 | 0.012 | 284.8 | 350 | 0.004 | **0.014** | **0.004** | **0.007** | **195.8** | **314.5** | **0.003** |
| Minusformer | **0.009** | 0.005 | 0.013 | 288.6 | **370.5** | 0.004 | 0.01 | **0.004** | **0.01** | **186.6** | 439 | **0.003** |
| WITRAN | **0.021** | 0.048 | 0.064 | 50.26 | 68.5 | **0.042** | 0.021 | **0.047** | **0.064** | **50.26** | **63.5** | 0.044 |
| Times2D | 0.139 | **0.237** | 0.24 | 915.2 | 1207 | **0.16** | **0.137** | 0.255 | **0.193** | **780.8** | **1077** | 0.234 |
| BiLSTM | 0.01 | 0.014 | 0.028 | 733.5 | 841 | 0.014 | **0.01** | **0.014** | **0.028** | **674** | **768** | **0.013** |
| ConvLSTM | 0.014 | 0.008 | **0.018** | 339.8 | 441 | 0.008 | **0.008** | **0.008** | 0.042 | **321.4** | **411** | **0.007** |
| GRU | 0.008 | 0.008 | **0.016** | 308.2 | 386.5 | 0.007 | **0.008** | **0.007** | 0.023 | **277.6** | **356.5** | **0.007** |
| LSTM | 0.007 | 0.008 | **0.018** | 324.6 | 409.5 | 0.008 | **0.007** | **0.007** | 0.018 | **288.1** | **374.5** | **0.007** |
| RNN | **0.008** | 0.008 | 0.025 | 154.5 | 208 | 0.007 | 0.008 | **0.007** | **0.012** | **137.3** | **175.5** | **0.007** |

Table 18: National illness forecasting results with and without embeddings for prediction horizons $H \in \{24, 36, 48, 60\}$. Bold values indicate better performance.

| Model | Metric | With Embedding | | | | Time | Mem | Without Embedding | | | | Time | Mem |
|---|---|---|---|---|---|---|---|---|---|---|---|---|---|
| | | H | | | | | | H | | | | | |
| | | 96 | 192 | 336 | 720 | | | 96 | 192 | 336 | 720 | | |
| PDF | MSE | **1.981** | **2.203** | **1.882** | **1.885** | 9.2 | 823 | 2.28 | 2.288 | 2.1 | 2.001 | **8.702** | **741** |
| | MAE | **0.842** | **0.873** | **0.838** | **0.869** | | | 0.903 | 0.935 | 0.894 | 0.91 | | |
| MICN | MSE | **2.809** | **2.83** | 2.918 | **2.91** | 4.775 | 738 | 2.879 | 2.862 | **2.849** | 2.965 | **3.552** | **563** |
| | MAE | **1.162** | 1.156 | 1.167 | **1.161** | | | 1.162 | **1.155** | **1.154** | 1.177 | | |
| ETSformer | MSE | **2.918** | **3.239** | **3.201** | **3.343** | 6.861 | 948 | 4.24 | 4.686 | 4.617 | 4.282 | **4.425** | **607** |
| | MAE | **1.174** | **1.237** | **1.212** | **1.232** | | | 1.442 | 1.53 | 1.51 | 1.44 | | |
| PatchTST | MSE | **1.785** | **1.677** | **1.587** | 2.252 | 4.798 | 1225 | 1.888 | 1.776 | 1.886 | **1.715** | **3.784** | **1083** |
| | MAE | **0.85** | **0.858** | **0.827** | 1.011 | | | 0.885 | 0.872 | 0.9 | **0.888** | | |
| SOFTS | MSE | 2.37 | **1.722** | 2.115 | 1.89 | 3.168 | 639 | **1.54** | 1.777 | **1.77** | 1.886 | **3.041** | **597** |
| | MAE | 0.9 | 0.865 | 0.914 | **0.925** | | | **0.751** | **0.848** | **0.871** | 0.938 | | |
| VarDrop | MSE | 3.004 | 2.708 | 2.576 | 2.285 | **3.493** | 1677 | **1.814** | **2.185** | **1.866** | **1.976** | 4.046 | **1632** |
| | MAE | 0.994 | 0.956 | 0.953 | 0.993 | | | **0.855** | **0.886** | **0.871** | **0.955** | | |
| Crossformer | MSE | 4.903 | 4.965 | **4.258** | **4.971** | 10.44 | 2585 | **4.583** | **4.925** | 4.47 | 5.069 | **10.35** | **2582** |
| | MAE | 1.545 | 1.542 | **1.383** | **1.561** | | | **1.493** | **1.541** | 1.447 | 1.584 | | |
| FlashAttention | MSE | 4.4 | 3.64 | 2.307 | **2.064** | 4.364 | 1355 | **1.521** | **2.466** | **2.086** | 2.067 | 4.697 | 1360 |
| | MAE | 1.069 | 1.053 | 0.966 | 0.981 | | | **0.796** | **0.941** | **0.92** | **0.979** | | |
| iFlowformer | MSE | 1.426 | **2.255** | 2.02 | 2.286 | 4.006 | 1163 | **1.398** | 2.299 | **1.931** | **1.925** | **3.51** | **1114** |
| | MAE | 0.794 | **0.905** | 0.916 | 1.017 | | | **0.761** | 0.911 | **0.907** | **0.935** | | |
| PPDformer | MSE | 3.785 | 2.444 | 2.378 | 3.809 | 9.519 | 2192 | **1.726** | **2.038** | **1.994** | **2.118** | **7.497** | **2014** |
| | MAE | 1.07 | 0.962 | 0.963 | 1.176 | | | **0.822** | **0.939** | **0.882** | **0.972** | | |
| LiNo | MSE | 1.791 | **1.729** | 1.958 | 2.345 | 3.713 | 567 | **1.744** | 1.883 | **1.904** | **1.959** | **3.225** | **545** |
| | MAE | 0.895 | 0.841 | 0.911 | 1.014 | | | **0.845** | **0.829** | **0.881** | 0.92 | | |
| EDformer | MSE | **2.531** | **3.053** | **2.87** | **2.849** | 2.268 | **1839** | 3.072 | 3.322 | 3.516 | 3.34 | **2.143** | 1970 |
| | MAE | **1.08** | **1.191** | **1.186** | **1.215** | | | 1.204 | 1.266 | 1.317 | 1.287 | | |
| Minusformer | MSE | 2.225 | 2.597 | 2.515 | 2.396 | 3.219 | 681 | **1.542** | **2.151** | **1.9** | **2.052** | **2.678** | **652** |
| | MAE | 0.925 | 0.94 | 0.935 | 0.934 | | | **0.788** | **0.868** | **0.84** | **0.906** | | |
| Times2D | MSE | 1.883 | 1.891 | 1.971 | 2.063 | 5.485 | 724 | **1.853** | **1.853** | **1.834** | 2 | **5.432** | **710** |
| | MAE | 0.84 | 0.871 | 0.909 | **0.931** | | | **0.819** | **0.859** | **0.874** | 0.945 | | |
| BiLSTM | MSE | 5.14 | 5.714 | **5.908** | **5.842** | 6.429 | 867 | **4.771** | **5.326** | 8.16 | 6.1 | **6.258** | **831** |
| | MAE | 1.496 | 1.606 | **1.653** | **1.651** | | | **1.43** | **1.558** | 2.072 | 1.679 | | |
| ConvLSTM | MSE | **5.403** | **5.635** | **5.841** | **5.939** | 4.409 | 673 | 5.674 | 6.229 | 6.427 | 6.202 | 4.419 | **667** |
| | MAE | **1.544** | **1.599** | **1.637** | **1.646** | | | 1.582 | 1.683 | 1.725 | 1.697 | | |
| GRU | MSE | 5.254 | 5.957 | **5.673** | 5.472 | 4.321 | 663 | **4.887** | **5.714** | 5.858 | **4.746** | 4.241 | **660** |
| | MAE | 1.514 | 1.641 | **1.593** | 1.591 | | | **1.466** | **1.611** | 1.641 | **1.495** | | |
| LSTM | MSE | 5.243 | 5.921 | **5.728** | **5.825** | 4.346 | 660 | **5.041** | **5.238** | 6.262 | 6.048 | **4.328** | **658** |
| | MAE | 1.524 | 1.634 | **1.626** | **1.656** | | | **1.486** | **1.536** | 1.69 | 1.659 | | |
| RNN | MSE | **4.937** | **5.568** | **5.12** | **5.226** | 2.68 | **611** | 4.98 | 5.664 | 5.254 | 5.439 | **2.351** | 620 |
| | MAE | 1.501 | **1.59** | **1.559** | **1.575** | | | **1.495** | 1.602 | 1.589 | 1.618 | | |

Table 19: Average efficiency results for the National Illness dataset with and without embeddings, including DataLoader time, forward pass time, backward pass with optimization time, peak allocated GPU memory, peak reserved GPU memory, and inference latency.

| Model | With Embedding | | | | | | Without Embedding | | | | | |
|---|---|---|---|---|---|---|---|---|---|---|---|---|
| | DL | FW | BW | PA | PR | Lat | DL | FW | BW | PA | PR | Lat |
| PDF | **0.041** | 0.042 | 0.064 | 352.2 | 388 | 0.035 | 0.045 | **0.041** | **0.06** | **224.3** | **243.5** | **0.023** |
| MICN | **0.015** | 0.005 | 0.013 | 165.9 | 393.5 | **0.004** | 0.015 | **0.004** | **0.006** | **54.38** | **76.5** | 0.004 |
| ETSformer | 0.015 | 0.012 | 0.025 | 633.3 | 753.5 | 0.022 | **0.014** | **0.007** | **0.01** | **118.5** | **124** | **0.006** |
| PatchTST | 0.008 | 0.004 | 0.008 | 152.1 | 158 | 0.003 | **0.007** | **0.002** | **0.006** | **152** | 158 | **0.002** |
| SOFTS | **0.012** | **0.002** | 0.006 | 201.5 | 222 | 0.002 | 0.013 | 0.002 | **0.005** | **156.8** | **184** | **0.002** |
| VarDrop | **0.012** | **0.003** | 0.01 | 224.6 | 266 | **0.003** | 0.013 | 0.005 | **0.009** | **158.3** | **186** | 0.004 |
| Crossformer | 0.009 | 0.008 | 0.023 | 836.1 | 897.5 | 0.006 | **0.009** | **0.008** | **0.023** | **834.2** | **891.5** | **0.006** |
| FlashAttention | **0.014** | **0.004** | **0.011** | 201.6 | 242.6 | **0.003** | 0.016 | 0.007 | 0.013 | **155.6** | **184** | 0.004 |
| iFlowformer | **0.013** | 0.004 | 0.01 | 219.4 | 248 | 0.003 | 0.013 | **0.004** | **0.007** | **156.1** | **184** | **0.003** |
| PPDformer | 0.016 | 0.024 | 0.029 | 613.2 | 759.5 | 0.022 | **0.015** | **0.021** | **0.018** | **245.5** | **611.5** | **0.02** |
| LiNo | 0.013 | 0.007 | 0.006 | 45.97 | 52 | 0.031 | **0.012** | **0.004** | **0.006** | **21.97** | **26.5** | **0.004** |
| EDformer | **0.044** | 0.004 | **0.007** | **133.7** | **168** | 0.003 | 0.044 | **0.004** | 0.009 | 187.3 | 438 | **0.002** |
| Minusformer | **0.012** | 0.003 | 0.008 | 229.3 | 272.5 | 0.003 | 0.012 | **0.003** | **0.005** | **164** | **245** | **0.002** |
| Times2D | 0.015 | 0.01 | 0.016 | 156.9 | 218 | 0.008 | **0.015** | **0.01** | **0.015** | **155.2** | **208** | **0.007** |
| BiLSTM | 0.017 | 0.012 | 0.023 | 319.5 | 390 | 0.012 | **0.016** | **0.012** | **0.022** | **299.4** | **360** | **0.011** |
| ConvLSTM | **0.014** | 0.007 | 0.014 | 146.4 | 180 | 0.007 | 0.014 | **0.006** | **0.013** | **140.3** | **176** | **0.006** |
| GRU | 0.015 | 0.007 | 0.013 | 138.7 | 172 | 0.006 | **0.015** | **0.006** | **0.013** | **126** | **168** | **0.006** |
| LSTM | 0.014 | 0.007 | 0.014 | 142.1 | 176 | 0.007 | **0.013** | **0.006** | **0.013** | **127** | **171** | **0.006** |
| RNN | 0.014 | 0.002 | 0.004 | 104.5 | 137.7 | 0.002 | **0.014** | **0.001** | **0.003** | **92.97** | **114.3** | **0.001** |

## A.5 Statistical significance analysis

Table 20: Confidence intervals for MSE of selected models on ETTh1 and ETTm1 with input length $L = 96$ and prediction horizons $H \in \{96, 192, 336, 720\}$. **Bold** values indicate better performance.

| Models | $H$ | Times2D | | PDF | | LiNo | | SOFTS | |
|---|---|---|---|---|---|---|---|---|---|
| | | **With Embedding** | | | | | | | |
| | | MSE | CI | MSE | CI | MSE | CI | MSE | CI |
| ETTh1 | 96 | 0.379 | (0.378, 0.380) | 0.385 | (0.382, 0.388) | 0.385 | (0.384, 0.386) | 0.384 | (0.383, 0.386) |
| | 192 | 0.431 | (0.429, 0.433) | 0.439 | (0.436, 0.442) | 0.442 | (0.438, 0.446) | 0.448 | (0.445, 0.450) |
| | 336 | 0.473 | (0.469, 0.476) | 0.492 | (0.486, 0.498) | 0.476 | (0.471, 0.480) | 0.501 | (0.494, 0.509) |
| | 720 | 0.473 | (0.469, 0.477) | 0.521 | (0.501, 0.544) | 0.482 | (0.473, 0.491) | 0.538 | (0.524, 0.552) |
| ETTm1 | 96 | 0.326 | (0.323, 0.328) | 0.335 | (0.334, 0.336) | 0.332 | (0.331, 0.333) | 0.328 | (0.325, 0.330) |
| | 192 | 0.371 | (0.370, 0.372) | 0.374 | (0.372, 0.376) | 0.383 | (0.375, 0.392) | 0.386 | (0.381, 0.391) |
| | 336 | 0.407 | (0.402, 0.411) | 0.403 | (0.401, 0.405) | 0.438 | (0.432, 0.444) | 0.438 | (0.428, 0.447) |
| | 720 | 0.459 | (0.455, 0.463) | 0.457 | (0.455, 0.459) | **0.496** | (**0.485**, 0.506) | 0.480 | (0.477, 0.482) |
| | | **Without Embedding** | | | | | | | |
| | | MSE | CI | MSE | CI | MSE | CI | MSE | CI |
| ETTh1 | 96 | **0.361** | (**0.359, 0.364**) | **0.378** | (**0.376, 0.380**) | **0.377** | (**0.375, 0.379**) | **0.383** | (**0.382, 0.384**) |
| | 192 | **0.428** | (**0.427, 0.429**) | **0.432** | (**0.428, 0.436**) | **0.428** | (**0.426, 0.429**) | **0.441** | (**0.438, 0.444**) |
| | 336 | **0.466** | (**0.463, 0.469**) | **0.479** | (**0.476, 0.482**) | **0.463** | (**0.460, 0.466**) | **0.487** | (**0.483, 0.490**) |
| | 720 | **0.472** | (**0.468, 0.476**) | **0.518** | (**0.499, 0.537**) | **0.470** | (**0.464, 0.476**) | **0.526** | (**0.514, 0.537**) |
| ETTm1 | 96 | **0.325** | (**0.322, 0.327**) | **0.324** | (**0.322, 0.326**) | **0.326** | (**0.323, 0.329**) | **0.322** | (**0.321, 0.323**) |
| | 192 | **0.369** | (**0.368, 0.370**) | **0.368** | (**0.365, 0.370**) | **0.373** | (**0.370, 0.376**) | **0.368** | (**0.367, 0.369**) |
| | 336 | **0.401** | (**0.397, 0.406**) | **0.395** | (**0.394, 0.397**) | **0.421** | (**0.415, 0.426**) | **0.406** | (**0.405, 0.407**) |
| | 720 | **0.455** | (**0.451, 0.459**) | **0.454** | (**0.452, 0.456**) | 0.497 | (0.492, **0.502**) | **0.476** | (**0.474, 0.477**) |

Table 21: Confidence intervals for MAE of selected models on ETTh1 and ETTm1 with input length $L = 96$ and prediction horizons $H \in \{96, 192, 336, 720\}$. **Bold** values indicate better performance.

| Models | $H$ | Times2D MAE | CI | PDF MAE | CI | LiNo MAE | CI | SOFTS MAE | CI |
|---|---|---|---|---|---|---|---|---|---|
| | | **With Embedding** | | | | | | | |
| | | MAE | CI | MAE | CI | MAE | CI | MAE | CI |
| ETTh1 | 96 | 0.402 | (0.400, 0.405) | 0.405 | (0.403, 0.407) | 0.403 | (0.402, 0.404) | 0.404 | (0.403, 0.405) |
| | 192 | 0.432 | (0.431, 0.433) | 0.438 | (0.436, 0.440) | 0.433 | (0.431, 0.436) | 0.442 | (0.440, 0.444) |
| | 336 | 0.442 | (0.440, 0.444) | 0.466 | (0.463, 0.470) | 0.446 | (0.444, 0.448) | 0.469 | (0.464, 0.475) |
| | 720 | 0.465 | (**0.462**, 0.468) | 0.497 | (0.484, 0.509) | 0.468 | (0.463, 0.473) | 0.513 | (0.504, 0.521) |
| ETTm1 | 96 | 0.364 | (0.362, 0.366) | 0.368 | (0.367, 0.369) | 0.367 | (0.366, 0.368) | 0.365 | (0.363, 0.366) |
| | 192 | 0.390 | (0.385, 0.395) | 0.392 | (0.390, 0.393) | 0.395 | (0.390, 0.399) | 0.397 | (0.394, 0.400) |
| | 336 | 0.410 | (0.405, 0.415) | 0.414 | (0.412, 0.416) | 0.426 | (0.423, 0.429) | 0.429 | (0.424, 0.433) |
| | 720 | 0.441 | (0.439, 0.443) | 0.465 | (0.457, 0.472) | **0.460** | (**0.455**, 0.465) | 0.455 | (0.454, 0.457) |
| | | **Without Embedding** | | | | | | | |
| | | MAE | CI | MAE | CI | MAE | CI | MAE | CI |
| ETTh1 | 96 | **0.392** | (**0.391**, **0.393**) | **0.399** | (**0.398**, **0.400**) | **0.399** | (**0.397**, **0.401**) | **0.402** | (**0.401**, **0.403**) |
| | 192 | **0.422** | (**0.421**, **0.423**) | **0.431** | (**0.429**, **0.433**) | **0.425** | (**0.424**, **0.426**) | **0.437** | (**0.435**, **0.439**) |
| | 336 | **0.439** | (**0.438**, **0.440**) | **0.453** | (**0.450**, **0.455**) | **0.440** | (**0.438**, **0.441**) | **0.463** | (**0.460**, **0.466**) |
| | 720 | 0.465 | (0.463, **0.467**) | **0.491** | (**0.481**, **0.500**) | **0.463** | (**0.459**, **0.466**) | **0.506** | (**0.500**, **0.513**) |
| ETTm1 | 96 | **0.363** | (**0.361**, **0.365**) | **0.362** | (**0.360**, **0.364**) | **0.363** | (**0.361**, **0.365**) | **0.361** | (**0.360**, **0.362**) |
| | 192 | **0.387** | (**0.385**, **0.389**) | **0.387** | (**0.385**, **0.388**) | **0.389** | (**0.387**, **0.390**) | **0.386** | (**0.385**, **0.387**) |
| | 336 | **0.406** | (**0.403**, **0.408**) | **0.408** | (**0.407**, **0.409**) | **0.419** | (**0.416**, **0.423**) | **0.411** | (**0.410**, **0.412**) |
| | 720 | **0.439** | (**0.438**, **0.441**) | **0.455** | (**0.451**, **0.460**) | 0.461 | (0.458, **0.464**) | **0.451** | (**0.451**, **0.452**) |

## A.6 Embedding and Input Dimensions Across Models

Table 22 summarizes the embedding dimensions used by each forecasting model when the data embedding layer is enabled, as well as the corresponding raw input dimension when embeddings are removed. For all datasets, the reported dimensions correspond to experiments with input sequence length $L = 96$ and prediction length $H = 96$, except for the National Illness dataset where a prediction horizon of $H = 60$ is used following standard practice. This table provides a unified view of how each model transforms input features under both settings, clarifying architectural differences across the full set of benchmark datasets.

Table 22: Embedding dimension $d_{\text{model}}$ (with embeddings) and input dimension $c_{\text{in}}$ (without embeddings) for all models. Values are for $L = 96$, $H = 96$, except National Illness ($H = 60$).

| Model | ETTh1 | | ETTh2 | | ETTm1 | | ETTm2 | | Weather | | Exchange | | National Illness | |
|---|---|---|---|---|---|---|---|---|---|---|---|---|---|---|
| | $d_{model}$ | $c_{in}$ | $d_{model}$ | $c_{in}$ | $d_{model}$ | $c_{in}$ | $d_{model}$ | $c_{in}$ | $d_{model}$ | $c_{in}$ | $d_{model}$ | $c_{in}$ | $d_{model}$ | $c_{in}$ |
| PDF | 512 | 7 | 512 | 7 | 512 | 7 | 512 | 7 | 64 | 21 | 64 | 8 | 64 | 7 |
| MICN | 512 | 7 | 512 | 7 | 512 | 7 | 512 | 7 | 32 | 21 | 512 | 8 | 64 | 7 |
| ETSformer | 512 | 7 | 512 | 7 | 512 | 7 | 512 | 7 | 512 | 21 | 512 | 8 | 512 | 7 |
| LiNo | 512 | 7 | 512 | 7 | 512 | 7 | 512 | 7 | 512 | 21 | 256 | 8 | 256 | 7 |
| Times2D | 64 | 7 | 64 | 7 | 64 | 7 | 64 | 7 | 64 | 21 | 64 | 8 | 64 | 7 |
| SOFTS | 256 | 7 | 128 | 7 | 128 | 7 | 256 | 7 | 512 | 21 | 512 | 8 | 512 | 7 |
| PatchTST | 512 | 7 | 512 | 7 | 512 | 7 | 512 | 7 | 128 | 21 | 512 | 8 | 16 | 7 |
| VarDrop | 512 | 7 | 512 | 7 | 512 | 7 | 512 | 7 | 512 | 21 | 512 | 8 | 512 | 7 |
| FlashAttention | 512 | 7 | 512 | 7 | 512 | 7 | 512 | 7 | 512 | 21 | 512 | 8 | 512 | 7 |
| iFlowformer | 512 | 7 | 512 | 7 | 512 | 7 | 512 | 7 | 512 | 21 | 512 | 8 | 512 | 7 |
| WITRAN | 32 | 7 | 32 | 7 | 32 | 7 | 32 | 7 | 32 | 21 | 32 | 8 | 32 | 7 |
| Minusformer | 512 | 7 | 512 | 7 | 512 | 7 | 512 | 7 | 512 | 21 | 512 | 8 | 512 | 7 |
| EDformer | 512 | 7 | 512 | 7 | 512 | 7 | 512 | 7 | 512 | 21 | 512 | 8 | 512 | 7 |
| PPDformer | 512 | 7 | 512 | 7 | 512 | 7 | 512 | 7 | 512 | 21 | 512 | 8 | 512 | 7 |
| Crossformer | 512 | 7 | 512 | 7 | 512 | 7 | 512 | 7 | 256 | 21 | 512 | 8 | 512 | 7 |

