# OpenReview forum: "Are Data Embeddings Effective in Time Series Forecasting?"
_TMLR — Accepted by TMLR_

### Review · Reviewer_Et3g · 2025-10-23

**Summary Of Contributions:**

The paper presents a systematic, model- and dataset-spanning ablation study on the role of *data embedding layers* in time-series forecasting. Concretely, the authors:

- Define and categorize common embedding mechanisms, including value, temporal, positional, inverted, and patch embeddings, and specify how each is removed while preserving shape compatibility.
- Evaluate 15 high-performing forecasting models across four ETT benchmarks under identical input length and horizon settings, recording accuracy (MSE/MAE) and efficiency (epoch time, memory).
- Report that *removing* embedding layers generally improves accuracy and reduces computational cost, often by margins larger than typical SOTA deltas reported in prior work. They also provide 95% confidence intervals showing gains are statistically reliable for selected models/horizons.
- Position the work as a cautionary, empirically grounded call to scrutinize default embedding usage in TS forecasting.

**Audience:**

Yes

**Audience Explanation:**

Readers working on time-series architectures, benchmarking, and efficient training would be keenly interested. The result challenges a widespread default design choice (embedding layers) across multiple paradigms (Transformers/MLPs/hybrids), with both accuracy and efficiency implications. Even researchers outside forecasting can draw lessons on rigorous component ablations and skepticism toward architectural boilerplate.

**Broader Impact Concerns:**

The authors articulate some potential concerns within their statement.
- **Risk of over-generalization:** If readers extrapolate beyond forecasting on ETT-like data, they might prematurely drop embeddings where they are beneficial (e.g., with rich exogenous variables, irregular sampling, or cross-domain transfer). The paper’s Limitations correctly cautions this, and strengthening guidance on “when *not* to remove” embeddings would mitigate misuse.
- **Benchmark lock-in:** Findings are tied to common ETT settings (*L=96*, horizons {96,192,336,720}); if the community narrows to these settings, models may be implicitly optimized for that regime. Encouraging broader datasets/tasks (classification, imputation) as the authors note is a healthy countermeasure.

**Claims And Evidence:**

Yes

**Claims Explanation:**

Overall, within the stated scope, the evidence is clear and convincing.

- **Clarity & methodology:** The paper carefully operationalizes “embedding removal” for each category with explicit shape conventions, which increases reproducibility and interpretability of the ablations.
- **Breadth of evidence:** The study spans 15 models and 4 standard datasets, with consistent preprocessing, horizons, and metrics. This breadth strengthens external validity within the ETT family.
- **Strength of results:** The core result on "accuracy often *improves* without embeddings" is repeatedly documented in Tables 2–3 and Appendix A.3 (Tables 8–9). The authors also contextualize the magnitudes against the tiny SOTA deltas in Table 1.
- **Statistical support:** 95% CIs for selected model/horizon pairs (Tables 4–5) show non-overlapping intervals in nearly all cases, supporting the claim of robust gains.
- **Nuanced discussion:** The paper explicitly calls out exceptions (e.g., EDformer) and offers plausible mechanisms (tiling/memory behavior; added permutations) rather than over-generalizing.

**Requested Changes:**

1. **Generalization beyond ETT (Limitations).**
   - Add at least one *non-ETT* dataset (e.g., traffic/energy/finance with different variable counts or noise regimes) or provide a small “stress test” suite (irregular sampling, covariate shift, missingness) to see if embedding removal still helps. If not feasible, more explicit guidance on when embeddings might still be valuable (e.g., very high-dimensional covariates, sparse exogenous signals) would be useful.
2. **Clarify removal protocol and invariances (Section 2).**
   - The paper states dimension-matching rules (e.g., set *d = N* or *d = L* after removal). Please formalize these invariances in a short lemma or checklist: what *exactly* is held fixed (parameter count, layer widths, receptive field) versus what is allowed to change. This will help others replicate across additional architectures.
3. **Reporting efficiency more fully (Section 5).**
   - Include wall-clock breakdowns (data loader vs forward/backward), peak vs reserved GPU memory, and inference-time results (throughput/latency) since practitioners often care more about inference than training.

---

> ### Author Response · Authors · 2025-10-31
> **Author Response to Reviewer Et3g**
>
> We sincerely thank the reviewer for the detailed and constructive feedback and appreciate the time and effort dedicated to evaluating our work. We have revised the paper accordingly:
>
> 1. Generalization beyond ETT:
>
> We expanded the experiments to include several non-ETT datasets with different resolutions and variable counts. Specifically:
> - Exchange Rate (8 variables, daily)
> - Weather (21 variables, hourly)
> - National Illness (7 variables, weekly).
>
> The results for these datasets are now shown in Tables 14, 16, and 18. Dataset descriptions are added in Appendix A4.
>
> 2. Clarifying the removal protocol:
>
> We improved Section 2 to clearly describe the exact rules we use after removing the embedding layer. We added a two detailed tables in Appendix A2 that summarizes these invariances by listing which architectural components change when embeddings are removed and which remain fixed. These tables provide a comprehensive checklist for replicating our removal protocol across different architectures and embedding types.
>
> 3. Reporting efficiency more fully:
>
> We have expanded the efficiency analysis. The revised Section 5 now includes:
> - wall-clock breakdowns (data loader, forward, backward),
> - peak vs reserved GPU memory,
> - inference latency and throughput.
>
> These metrics are reported for each model in Tables 3, 5, 11, 13, 15, 17, and 19. We believe this addresses the concern and provides a clearer view of practical performance for both training and inference.

---

### Review · Reviewer_oTsL · 2025-10-29

**Summary Of Contributions:**

### Summary
The paper examines the role and necessity of data embedding layers in modern time series forecasting models. These layers, which project raw inputs into higher-dimensional spaces, are widely used but rarely justified. Through extensive ablation studies on fifteen state-of-the-art models and four benchmark datasets, the authors assess the impact of removing these layers. Interestingly, the results show that excluding data embeddings generally does not harm forecasting accuracy and often improves both performance and computational efficiency.

### Strengths
- The research question is clear, relevant, and addresses a common but underexplored design choice.
- The experimental evaluation is comprehensive, covering a diverse set of models and datasets.

### Weaknesses
- It appears that the embedding dimension is always set equal to the input (or patch, etc.) size. However, in practice, embeddings are often larger than the input dimension to expand the feature space, which may contribute to improved performance. By fixing the embedding size equal to the input, the layer reduces to a simple linear transformation, which limits the comparison’s validity. Including experiments with larger embedding dimensions is required for a meaningful evaluation.

**Audience:**

Yes

**Audience Explanation:**

The paper addresses a practical and widely relevant question in time series forecasting: whether data embedding layers, a common architectural component, are actually necessary. Since many recent models in this field adopt embeddings by default, understanding their true impact is valuable for both researchers and practitioners. With the right experimental setup (see above), this could give interesting insights.

**Claims And Evidence:**

No

**Claims Explanation:**

While the experiments are extensive and the results are clearly presented, the evidence provided does not fully support the paper’s broader claims. The study evaluates only cases where the embedding size equals the input dimension, which does not reflect typical practice where embeddings often expand the feature space. Without experiments testing larger embedding dimensions, it is unclear whether the conclusions generalize beyond this restricted setting. Therefore, the evidence is not yet sufficient to convincingly support the main claim.

**Requested Changes:**

1. Expand the embedding dimension analysis: Include experiments where the embedding size is larger than the input dimension (e.g., following the setup of the original models). This is essential to evaluate whether the observed results hold when the feature space is expanded, as is common in practice.
2. Discuss the role of the embedding dimension: Add a discussion where you analyze the role of the embedding size and how it influences the overall model performance.

---

> ### Author Response · Authors · 2025-10-31
> **Author Response to Reviewer oTsL**
>
> We sincerely thank the reviewer for the detailed and constructive feedback and appreciate the time and effort dedicated to evaluating our work. We would like to clarify an important point regarding the experimental setup, particularly regarding embedding dimensionality.
>
> In the baseline configurations (with embeddings), we strictly followed the original model implementations as released by their authors. These implementations use expanded embedding dimensions that are larger than the input size. For example, LiNo, ETSformer, and PDF use an embedding dimension of 512, while Times2D uses 64. This means that the embedding layers in our baseline models indeed project the raw inputs into higher-dimensional spaces. In the embedding-removed scenario, we eliminated the embedding projection layer entirely. However, the subsequent layers in each model were originally designed to receive inputs with the embedding dimension (e.g., 512 or 1024). To preserve architectural compatibility after removing the embedding, we directly passed the raw input to these layers. That is why we mentioned in the paper in scenarios where embedding is not used embedding size is set the input dimension.
>
> To clarify this setup, we made three revisions in the updated manuscript:
>
> - Section 2 (“Data embeddings and their removal”) now explicitly explains how input dimensions are routed through models after removing embeddings.
> - Appendix A.2 now includes two detailed tables that summarize which architectural components change when embeddings are removed and which remain fixed when the feature or temporal dimension are transformed into the embedding dimension.
> - Appendix A.6 now includes a table summarizing the actual embedding sizes used for all models and datasets.
>
> 2. Role of the embedding dimension:
> Since the embedding size is typically greater than the input dimension in all baseline models (as shown in Table A5), all accuracy and efficiency results and discussions throughout the paper reflect experiments with expanded feature spaces. We acknowledge that a comprehensive evaluation of multiple embedding sizes (e.g., 64, 128, 256, 512, 1024, 2048) across all 15 models and 7 datasets would provide additional insights. However, such a full exploration is beyond the scope of this study, as it would require retraining over a thousand additional model configurations. The embedding sizes are hyperparameters of the original models, and we made a deliberate effort to use the same embedding sizes from the original implementations to ensure fair comparison and reproducibility. Our primary contribution is demonstrating that the commonly used embedding techniques in modern state-of-the-art models may not be necessary for achieving competitive performance.

---

> > ### Comment · Reviewer_oTsL · 2025-12-11
> >
> > Dear authors, thank you for the detailed clarification, and apologies for the earlier misinterpretation. I appreciate the effort in clarifying the details on the embedding removal in the paper. With this clarification, my concerns are addressed.

---

### Review · Reviewer_ihYN · 2025-11-07

**Summary Of Contributions:**

Time series forecasting is one of the core use case of machine learning, across industries. The paper claims that applying state-of-the art machine learning architectures have only improved marginally, frequently only by a thousandth of a point. In fact, removal of removal of embeddings could result in better forecasting performance. The main focus is the usage of data embedding techniques.

The authors removed the embeddings layer from the algorithms to compare the performance.The choices made to run the models without embeddings are reasonable.

The related works sections covers a wide variety of forecasting models. It includes MICN, SOFTS, Times2D to name a few and explains their approach to forecasting.

The data is comprehensive and clearly shows the improvements in MAE, MSE, as well as compute performance. Limitations correctly call out the need to look into downstream impact of removing embeddings, and investigating the impact on other real time datasets.

**Audience:**

Yes

**Audience Explanation:**

With the advent of machine learning and AI capabilities, everyone is trying to implement these state of the art technologies across industrial applications. Embeddings is a core part of these state-of-the-art technologies. However, if embeddings are only adding complexity and cost, without improving the performance, in face, reducing the performance in some cases, then it's not worth the effort, complexity, and cost.

**Broader Impact Concerns:**

None, the broader impact section rightly calls out that focusing on reanalyzing existing architectures may result in better economic value.

**Claims And Evidence:**

Yes

**Claims Explanation:**

Yes, the authors provide comprehensive details of the methodology and results:
- Explained different types of embeddings and substitute values when embeddings ere not present
- Explained the overall methodology of running the original models and then rerunning without embeddings to have a comparable benchmark
- The results provided clearly support the claims made in the paper. The methodology is explained well, and the data presented is comprehensive to establish superiority of performance without embeddings layer. Especially, on longer term horizons, the difference is significant
- The authors also compare the impact on compute performance. As expected, models without embeddings performed better here. But I appreciate that this part was included
- The results hold across data sets, and models. Although, more work will be needed to verify that the results hold on real time data sets across different industries. This has been called out as a limitation of the work.

**Requested Changes:**

Table 1: The data includes only state-of-the art models so it is not surprising that the performance is close. It'd have helped to include here benchmarking against models that don't include embeddings or are generally simpler than the complex ML implementations to see the true difference.

Related works: Why were only these specific forecasting models included in the comparison and in the related works. It'd be helpful to include whether these cover all the noteworthy approaches or if there are some left out. Essentially, can the findings on the selected models representative of all the models that use embeddings, or are there cases which are not covered in the paper but may perform differently due to fundamental differences in how those would use embeddings.

Minor suggestion: Instead of having a separate table for statistical significance, it may be more efficient to denote in tables 2 and 3, which values are statistically significant. Table 4 can move in appendix.

It would help to include some background on the datasets used, and why they are relevant and sufficient. Summarizing the details of the split will also help

---

> ### Author Response · Authors · 2025-12-01
> **Author Response to Reviewer ihYN**
>
> We sincerely thank the reviewer for the detailed and constructive feedback and appreciate the time and effort dedicated to evaluating our work. We have revised the paper accordingly:
>
> 1. Table 1 - Benchmarking against simpler models:
> We agree that including simpler models without embedding layers can help reveal the true performance differences. Table 1 focuses exclusively on recent state-of-the-art models to illustrate the minimal performance gaps among them, and was limited by space constraints in the original submission. Therefore, we have added five simpler benchmarks—RNN, LSTM, GRU, ConvLSTM, and BiLSTM—across all datasets in the main and appendix tables. These models serve as embedding-free baselines and listing them next to the 15 state-of-the-art models helps to better highlight the true performance differences.
>
> 2. Related works:
> Our paper evaluates the most common embedding techniques in modern time series forecasting. To this end, we made a deliberate effort to select models with diverse architectures—Transformer-based, MLP-based, and hybrid/decomposition-based models—plus five traditional methods for a more comprehensive comparison. These models represent high-impact architectures with reproducible public code. We acknowledge that our findings may not cover every possible forecasting architecture. However, the embedding mechanisms we analyze are among the most widely used in time series forecasting, so we believe the conclusions are broadly representative. Our goal is to encourage researchers to critically assess the role of embedding layers, even when designing models with fundamentally different embedding mechanisms.
>
> 3. Statistical significance presentation:
> We have moved the statistical significance table to the appendix A5.
>
> 4. Dataset background and details:
> We have added additional background information on the datasets, including their relevance for our study. These additions are included in Appendix A.3.

---

### Decision · Action_Editor_kdkK · 2025-12-21

**Recommendation:** Accept with minor revision

**Audience:**

Yes

**Audience Explanation:**

I think a wide range of TMLR audience will be interested in this work and the results can also extend beyond time series forecasting models.

**Claims And Evidence:**

Yes

**Claims Explanation:**

Time series forecasting is one of the most important areas within machine learning and the work asks an important and pertaining question: are embeddings actually useful in the task of time series forecasting. Although "effective" is an overloaded term, the authors mean effectiveness in terms of performance. The authors perform extensive ablation studies on 15 state-of-the-art models and 4 benchmark datasets to assess the impact of the embedding layers within these models. Although there are no theorertical insights, the paper puts its message forward convincingly.

The initial reviews raised several concerns, the most pressing of which were:

* Lack of simpler baselines. I agree with the reviewer that including simpler ML models would give a more holistic picture of the effectiveness of the embedding based models.

* Altering the size of the embeddings was also an important point that was raised.

* Extending the experiments beyond the ETT benchmarks.

The authors gave a detailed rebuttal and at the end, the reviewers were in favor of accepting the paper. I agree with their assessment and believe that the work can give new insights to the time series forecasting practitioners.

I would request the authors to take the comments and rebuttals into account when preparing the camera ready version of the paper. Congratulations on the acceptance.